# Lévy-noise versus Gaussian-noise-induced Transitions in the Ghil-Sellers Energy Balance Model

Valerio Lucarini[1,2], Larissa Serdukova[1,2], and Georgios Margazoglou[1,2]

[1]Department of Mathematics and Statistics, University of Reading, Reading, UK.
[2]Centre for the Mathematics of Planet Earth, University of Reading, Reading, UK.

**Correspondence:** Valerio Lucarini (v.lucarini@reading.ac.uk)

**Abstract.** We study the impact of applying stochastic forcing to the Ghil-Sellers energy balance climate model in the form of a fluctuating solar irradiance. Through numerical simulations, we explore the noise-induced transitions between the competing warm and snowball climate states. We consider multiplicative stochastic forcing driven by Gaussian and $\alpha$-stable Lévy - $\alpha \in (0, 2)$ - noise laws, examine the statistics of transition times, and estimate most probable transition paths. While the Gaussian noise case - used here as a reference - has been carefully studied in a plethora of investigations on metastable systems, much less is known about the Lévy case, both in terms of mathematical theory and heuristics, especially in the case of high- and infinite-dimensional systems. In the weak noise limit, the expected residence time in each metastable state scales in a fundamentally different way in the Gaussian vs. Lévy noise case with respect to the intensity of the noise. In the former case, the classical Kramers-like exponential law is recovered. In the latter case, power laws are found, with the exponent equal to $-\alpha$, in apparent agreement with rigorous results obtained for additive noise in a related - yet different - reaction-diffusion equation as well as in simpler models. This can be better understood by treating the Lévy noise as a compound Poisson process. The transition paths are studied in a projection of the state space and remarkable differences are observed between the two different types of noise. The snowball-to-warm and the warm-to-snowball most probable transition path cross at the single unstable edge state on the basin boundary. In the case of Lévy noise, the most probable transition paths in the two directions are wholly separated, as transitions apparently take place via the closest basin boundary region to the outgoing attractor. This property can be better elucidated by considering singular perturbations to the solar irradiance.

## 1 Introduction

### 1.1 Multistability of the Earth's Climate

The climate system comprises five interacting subdomains: the atmosphere, the hydrosphere (water in liquid form), the upper layer of the lithosphere, the cryosphere (water in solid form), and the biosphere (ecosystems and living organisms). The climate is driven by the inhomogeneous absorption of incoming solar radiation, which sets up nonequilibrium conditions. The system reaches an approximate steady state where macroscopic fluxes of energy, momentum, and mass are present throughtout its domain, and entropy is continuously generated and expelled into the outer space. The climate features variability on a vast range of spatial and temporal scales as a result of the interplay of forcing, dissipation, feedbacks, mixing, transport, chemical

reactions, phase changes, and exchange processes between the subdomains; see Peixoto and Oort (1992) Lucarini et al. (2014a), Ghil (2015), and Ghil and Lucarini (2020).

In the late 1960s Budyko (1969) and Sellers (1969) independently proposed that in the current astronomical and astrophysical configuration the Earth could support two distinct climates, the present day Warm (W) state, and a competing one characterised by global glaciation, usually referred to as the Snowball (SB) state. Their analysis was performed using one-dimensional energy balance models (EBM)s, which, despite their simplicity, were able to capture the essential physical mechanism in action, i.e. the interplay between two key feedbacks. The Boltzmann feedback is associated with the fact that warmer bodies emit more radiation, and is a negative, stabilizing one. Instead, the instability of the system is due to the presence of the so-called ice-albedo feedback: an increase in the ice-covered fraction of the surface leads to further temperature reduction of the planet because ice reflects efficiently the incoming solar radiation. These mechanisms are active at all spatial scales, including the planetary one; see Budyko (1969) and Sellers (1969). Such pioneering investigations of the multistability of the Earth's climate were later extended by Ghil (1976) - see also the later analysis by Ghil and Childress (1987) - who provided a comprehensive mathematical framework for the problem based on the study of the bifurcations of the system. The main control parameter defining the stability properties is the solar irradiance $S^*$. Below the critical value $S_{W \to SB}$, only the snowball state is permitted, whereas above the critical value $S_{SB \to W}$, only the warm state is permitted. Such critical values, which determine the region of bistability, are defined by bifurcations that emerge when, roughly speaking, the strength of the positive, destabilising feedbacks becomes as strong as the negative, stabilizing feedbacks. Many variants of the models proposed by Budyko and Sellers have been discussed in the literature, all featuring by and large rather similar qualitative and quantitative features (Ghil, 1981; North et al., 1981; North, 1990; North and Stevens, 2006). Furthermore, these models have long been receiving a great deal of attention from the mathematical community regarding the possibility of proving existence of solutions and evaluating their multiplicity (Hetzer, 1990; Díaz et al., 1997; Kaper and Engler, 2013; Bensid and Díaz, 2019).

Only later these predictions were confirmed by actual data. Indeed, geological and paleomagnetic evidence suggests that during the Neoproterozoic era, between 630 and 715 million years ago, the Earth went at least twice into major long-lasting global glaciations that can be associated with the SB state; see Pierrehumbert et al. (2011) and Hoffman et al. (1998). Multicellular life emerged in our planet shortly after the final deglaciation from the last SB state (Gould, 1989). The robustness and importance of the competition between the Bolzmann feedback and the ice-albedo feedback in defining the global stability properties of the climate has been confirmed by investigations performed using higher complexity models (Lucarini et al., 2010; Pierrehumbert et al., 2011), including fully coupled climate models (Voigt and Marotzke, 2010). While the mechanisms described above are pretty robust, the concentration of greenhouse gases as well as the boundary conditions defined by the extent and position of the continents have an impact on the values of $S_{W \to SB}$ and $S_{SB \to W}$ as well as on the the properties of the competing states. The presence of multistability has a key importance in terms of determining habitability conditions for Earth-like exoplanets; see Lucarini et al. (2013) and Linsenmeier et al. (2015).

Additionally, several results indicate that the phase space of the climate system might well be more complex than the scenario of bistability described above. Various studies (Lewis et al., 2007; Abbot et al., 2011; Lucarini and Bódai, 2017; Margazoglou et al., 2021) performed with highly nontrivial climate models report the possible existence of additional competing states, up

to a total of five (Brunetti et al., 2019; Ragon et al., 2021). In Margazoglou et al. (2021) it is argued that, in fact, one can see the climate as a multistable system where multistability is realised at different hierarchical levels. As an example, the tipping points (Lenton et al., 2008; Steffen et al., 2018) that characterise the current (W) climate state can be seen as a manifestation of a hierarchically lower multistability with respect to the one defining the dichotomy between the W and SB states.

## 1.2 Transitions between Competing Metastable States: Gaussian vs Lévy Noise

Clearly, in the case of autonomous systems where the phase space is partitioned in more than one basin of attraction of the corresponding attractors and the basin boundaries, the asymptotic state of the system is determined by its initial conditions. Things change dramatically when one includes time-dependent forcing which allows for transitions between competing metastable states (Ashwin et al., 2012). In particular, following the viewpoint originally proposed by Hasselmann (1976), whereby the fast variables of the climate system act as stochastic forcings for the slow ones (Imkeller and von Storch, 2001), the relevance

of studying noise-induced transitions between competing states has become apparent (Hänggi, 1986; Freidlin and Wentzell, 1984). This viewpoint, where the noise is usually assumed to be Gaussian distributed, has provided very fruitful insight on the multiscale nature of climatic time series (Saltzman, 2001), and is related to the discovery of phenomena like stochastic resonance (Benzi et al., 1981; Nicolis, 1982).

Metastability is ubiquitous in nature and advancing its understanding is a key challenge in complex system science at large

(Feudel et al., 2018). In general, the transitions between competing metastable states in stochastically perturbed multistable systems take place, in the weak noise limit, through special regions of the basin boundaries, named edge states. The edge states are saddles: trajectories initialised in the basin boundaries are attracted to them, but there is an extra direction of instability, so that a small perturbation sends an orbit towards one of the competing metastable states with probability one (Grebogi et al., 1983; Ott, 2002; Kraut and Feudel, 2002; Skufca et al., 2006; Vollmer et al., 2009). In the case the edge state supports

chaotic dynamics, we refer to it as Melancholia (M) state (Lucarini and Bódai, 2017). In previous papers, we have shown that it is possible to construct M states in high-dimensional climate models (Lucarini and Bódai, 2017) and to prove that the nonequilibrium quasi-potential formalism introduced by Graham (1987) and Graham et al. (1991) provides a powerful framework for explaining the population of each metastable state and the statistics of the noise-induced transitions. In the weak-noise limit, edge states act as gateways for noise-induced transitions between the metastable states (Lucarini and Bódai,

2019; Lucarini and Bódai, 2020; Margazoglou et al., 2021); see also a recent study on a nontrivial metastable prey-predator model (Garain and Sarathi Mandal, 2022). The local minima and the saddles of the quasi-potential $\Phi$, which generalises the classical energy landscape for non-gradient systems, correspond to competing metastable states and to edge states, respectively. In our investigation, the climate system is forced by adding a random - Gaussian distributed - component to the solar irradiance, which impacts, in the form of multiplicative noise, only a small subset of the degrees of freedom of the system. We remark

that such choice of the stochastic forcing does not fully reflect physical realism, as the variability of the solar irradiance has a more complex behaviour (Solanki et al., 2013). Instead, noise acts as a tool for exploring the global stability properties of the system, and injecting noise as fluctuation of the solar irradiance has the merit of impacting the Lorenz energy cycle, thus

effecting all degrees of freedom of the system (Lucarini and Bódai, 2020). See also the recent detailed mathematical analysis of the stochastically perturbed one-dimensional EBMs presented in Díaz and Díaz (2021).

A major limitation of this mathematical framework is the need to rigidly consider Gaussian noise laws, even if considerable freedom is left as to the choice of the spatial correlation properties of the noise. It seems natural to attempt a generalization by considering the whole class of $\alpha$-stable Lévy noise laws. Lévy processes (Applebaum, 2009; Duan, 2015), described in detail below in Appendix A are fundamentally characterised by the stability parameter $\alpha \in (0,2]$, where the $\alpha = 2$ case corresponds to the Gaussian case (which is, indeed, a special Lévy process). In what follows, when we discuss Lévy noise laws, we refer to

$\alpha \in (0,2)$.

     Note that $\alpha$-stable Lévy processes have played an important role in geophysics as they have provided the starting point for defining the multiplicative cascades also referred to as *universal multifractal*. This framework has been proposed as way to analyze and simulate at climate scales the ubiquitous intermittency and heavy-tailed statistics of clouds (Schertzer and Lovejoy, 1988), rain reflectivity (Tessier et al., 1993; Schertzer and Lovejoy, 1997), atmospheric turbulence (Schmitt et al., 1996), and

soil moisture (Millàn et al., 2016). On longer time scales, multiplicative cascades have been used to interpret temperature records in the Summit ice core Schmitt et al. (1995); see Lovejoy and Schertzer (2013) for a summary of this view point. Mathematicians, on the other hand, have defined a *Lévy multiplicative chaos* (Fan, 1997; Rhodes et al., 2014) as a more mathematically tractable alternative to the universal multifractal. Finally, we remark that fractional Fokker-Planck equations have been proposed by Schertzer et al. (2001) to investigate the properties of nonlinear Langevin-type equations forced by a

$\alpha-$ stable Lévy noise with the goal of analysing and simulating anomalous diffusion.

     Following Ditlevsen (1999), it has become apparent that more general classes of $\alpha$-stable Lévy noise laws might be useful for modelling noise-induced transitions in the climate system like Dansgaard–Oeschger events. The viewpoint by Ditlevsen was particularly effective in stimulating mathematical investigations into noise-induced escapes from attractors where as stochastic forcing one chooses a Lévy, rather than Gaussian, noise (Imkeller and Pavlyukevich, 2006a, b; Chechkin et al., 2007; Debussche

et al., 2013). Such analyses have clarified that a fundamental dichotomy exists with the classical Freidlin and Wentzell scenario mentioned above, even if phenomena like stochastic resonance can be recovered also in this case (Dybiec and Gudowska-Nowak, 2009; Kuhwald and Pavlyukevich, 2016). Whereas in the Gaussian case transitions between competing attractors occur as a result of the very unlikely combination of many steps all going in the *right* direction, in the Lévy case, transitions result from individual, very large and very rare jumps. Recently, Duan and collaborators have made fundamental progresses in

achieving a variational formulation of Lévy noise-perturbed dynamical systems (Hu and Duan, 2020) as well as in developing corresponding methods for data assimilation (Gao et al., 2016) and data analysis (Lu and Duan, 2020). In terms of applications, Lévy noise is becoming a more and more a popular concept and tool for studying and interpreting complex systems (Grigoriu and Samorodnitsky, 2003; Penland and Sardeshmukh, 2012; Zheng et al., 2016; Wu et al., 2017; Serdukova et al., 2017; Cai et al., 2017; Singla and Parthasarathy, 2020; Gottwald, 2021).

The contribution by Gottwald (2021) is especially worth recapitulating because of its methodological clarity. There, the idea is, following Ditlevsen (1999), to provide a conceptual deterministic climate model able to generate a Lévy-noise-like signal to describe, at least qualitatively, abrupt climate changes similar to Dansgaard–Oeschger events, which are sequences of periods

of abrupt warming followed by slower cooling that occurred during the last glacial period (Barker et al., 2011). A key building block is the idea proposed in Thompson et al. (2017) that a Lévy noise can be produced by integrating the so-called correlated additive and multiplicative (CAM) noise processes, which are defined starting from standard Gaussian processes. The other key ingredient is to consider the atmosphere as the fast component in the multiscale model and deduce, using homogeneization theory (Pavliotis and Stuart, 2008; Gottwald and Melbourne, 2013), that its influence on the slower climate components can be closely represented as a Gaussian forcing. Finally, the temperature signal is cast as the integral of a CAM process.

We remark that Gaussian and Lévy noise can be associated with stochastic forcings of fundamentally different nature. One might think of Gaussian noise as being associated to the impact of very rapid unresolved scales of motion on the resolved ones Pavliotis and Stuart (2008). Instead, one might interpret $\alpha$-stable Lévy noise as describing, succinctly, the impact of what in the insurance sector are called acts of God (e.g. an asteroid hitting the Earth; a massive volcanic eruption; the sudden collapse of the West Antarctic ice sheet).

## 1.3  Outline of the Paper and Main Results

We consider here the Ghil-Sellers Earth's EBM Ghil (1976), a diffusive one-dimensional energy balance system, governed by a nonlinear reaction-diffusion parabolic partial differential equation. We stochastically perturb the system by adding random fluctuations to the solar irradiance, therefore the noise is introduced in multiplicative form. We study the transitions between the two competing metastable climate states and carry out a comparison of the effect of considering Lévy vs Gaussian noise laws of weak intensity $\varepsilon$.

The main challenges of the problem are: a) the fact that we are considering dynamical processes occurring in infinite dimensions (Doering, 1987; Duan and Wang, 2014; Alharbi, 2021); and b) the consideration of multiplicative Lévy noise laws (Peszat and Zabczyk, 2007; Debussche et al., 2013). We characterize noise-induced transitions between the competing climate basins and quantify the effect of noise parameters on them by estimating the statistics of escape times and empirically constructing mean transition pathways called instantons.

The results obtained confirm that, in the weak noise limit $\varepsilon \to 0$, the mean residence time in each metastable state driven by Gaussian vs. Lévy noise has a fundamentally different dependence on $\varepsilon$. Indeed, as expected, in the Gaussian case the residence time grows exponentially with $\varepsilon^{-2}$, thus in basic agreement with the well-known Kramers (1940) lsaw and the previous studies performed on climate models (Lucarini and Bódai, 2019; Lucarini and Bódai, 2020). Instead, in the case of $\alpha$-stable noise laws, the residence time increases with $\varepsilon^{-\alpha}$. We perform simulations for $\alpha = \{0.5, 1.0, 1.5\}$. The obtained scaling can be explained by treating effectively the Lévy noise as compound Poisson process and is in agreement with what is found for low dimensional dynamics (Imkeller and Pavlyukevich, 2006a, b), as well as with the infinite dimensional stochastic Chafee-Infante reaction-diffusion equation (Debussche et al., 2013) in the case of additive noise. This might indicate that such scaling laws are more general than what typically considered.

Furthermore, we find clear confirmation that, in the case of Gaussian noise in the weak noise limit, the escape from either attractor's basin takes place through the edge state. Indeed, the most probable paths for both thawing and freezing processes meet at the edge state and have distinct instantonic and relaxation sections. In turn, for Lévy noise in the weak-noise limit, the

escapes from a given basin of attraction occur through the boundary region closest to the outgoing attractor. Hence, the paths are very different from the Gaussian case (especially so for the freezing transition) and, somewhat surprisingly, are identical regardless the value of $\alpha$ considered. These properties can be better understood by studying the impact of including singular
perturbations to the value of the solar irradiance.

The rest of the paper is organized as follows. In Section 2 we present the Ghil-Sellers EBM and summarize its most important dynamical aspects, as well as the steady-state solutions and their stability. The stochastic partial differential equation obtained by randomly perturbing the solar irradiance in the EBM is given in Subsection 3.1, where we also clarify the mathematical meaning of the solution of the stochastic partial differential equation. Subsection 3.2 introduces the mean residence time and
most probable transition path between the competing climate states. The numerical methods are also briefly presented. In Section 4 we discuss our main results. In Section 5 we present our conclusions and perspectives for future investigations. Finally, Appendix A presents a succinct description of $\alpha$-stable Lévy processes, Appendix B sketches the derivation of the scaling laws for mean residence times presented in Debussche et al. (2013), Appendix C explores the behavior and dynamics of singular Lévy perturbations of different duration, and Appendix D presents a tabular summary of the statistics of the problem.

## 2  The Ghil-Sellers energy balance climate model

The Ghil-Sellers EBM (Ghil, 1976) is described by an one-dimensional nonlinear, parabolic, reaction-diffusion partial differential equation (PDE) (1) involving the surface temperature $T$ field and the transformed space variable $x = 2\phi/\pi \in [-1,1]$, where $\phi \in [-\pi/2, \pi/2]$ is the latitude. The model describes the processes of energy input, output, and diffusion across the domain and can be written as:

$$C(x)T_t = D_I(x,T,T_x,T_{xx}) + D_{II}(x,T) - D_{III}(T),\tag{1}$$

where $C(x)$ is the effective heat capacity and $T = T(x,t)$ has boundary and initial conditions as follows

$$T_x(-1,t) = T_x(1,t) = 0, \qquad T(x,0) = T_0(x).\tag{2}$$

The equation does not depend explicitly on the time $t$. The subscripts $_t$ and $_x$ refer to partial differentiation. The first term - $D_I$ - on the right hand side of (1) can be written as

$$D_I(x,T,T_x,T_{xx}) = \frac{4}{\pi^2 \cos(\pi x/2)}[\cos(\pi x/2)K(x,T)T_x]_x\tag{3}$$

and describes the convergence of meridional heat transport performed by the geophysical fluids. The function $K(x,T)$ is a combined diffusion coefficient, given by

$$K(x,T) = k_1(x) + k_2(x)g(T), \ \text{with}\tag{4}$$
$$g(T) = \frac{c_4}{T^2}\exp\left(-\frac{c_5}{T}\right).\tag{5}$$

The empirical functions $k_1(x)$ and $k_2(x)$ are eddy diffusivities for sensible and latent heat, respectively, and $g(T)$ is associated with the Clausius-Clapeyron relation, which describes the relationship between temperature and saturation water vapour content of the atmosphere.

     The second term - $D_{II}$ - of (1) describes the energy input associated with the absorption of incoming solar radiation can be written as

$$D_{II}(x,T) = \mu \mathcal{Q}(x)[1 - \alpha_a(x,T)], \tag{6}$$

where $\mathcal{Q}(x)$ is the incoming solar radiation and $\alpha_a(x,T)$ is the surface reflectivity (albedo), which is expressed as

$$\alpha_a(x,T) = \{b(x) - c_1(T_m + \min[T - c_2 z(x) - T_m,\, 0])\}_c, \tag{7}$$

where the subscript $\{\cdot\}_c$ denotes a cutoff for a generic quantity $h$ defined as

$$h_c = \begin{cases} h_{min} & h \le h_{min}, \\ h & h_{min} < h < h_{max}, \\ h_{max} & h_{max} \le h. \end{cases} \tag{8}$$

The term $c_2 z(x)$ in (7) indicates the difference between the sea-level and surface-level temperatures, and $b(x)$ is a temperature independent empirical function of the albedo. The parametrization given in Eqs. (7)-(8) encodes the positive ice-albedo feedback. The relative intensity of the solar radiation in the model can be controlled by the parameter $\mu$.

     The last term - $D_{III}$ - of (1) describes the energy loss to space by outgoing thermal planetary radiation and is responsible for the negative Boltzmann feedback. It can be written as

$$D_{III}(T) = \sigma T^4[1 - m \tanh(c_3 T^6) \tag{9}$$

where $\sigma$ is the Stefan-Boltzmann constant and the emissivity coefficient is expressed as $1 - m \tanh(c_3 T^6)$. Such term describes, in a simple yet effective way, the greenhouse effect by reducing infrared radiation losses. The values of the empirical functions $C(x), Q(x), b(x), z(x), k_1(x), k_2(x)$ at discrete latitudes and empirical parameters $c_1, c_2, c_3, c_4, c_5, \sigma, m, T_m$ are taken from Ghil (1976), as modified in Bódai et al. (2015). The choice of the empirical functions and parameters are extensively discussed

in Ghil (1976). Of course one might reasonably wonder about the robustness of our modelling strategy. Indeed, a plethora of EBMs analogous to the one described here have been presented in the literature, where slightly different parametrizations for the diffusion operator, for the albedo, and for the greenhouse effect are introduced. Such models are in fundamental agreement both in terms of physical (Ghil, 1981; North et al., 1981; North, 1990; North and Stevens, 2006) and mathematical properties (Hetzer, 1990; Díaz et al., 1997; Kaper and Engler, 2013; Bensid and Díaz, 2019)

In this study, we consider $\mu = 1.05$. For this value of $\mu$, two stable asymptotic states - the W and the SB states - co-exist, see Figure 1b, reproduced from Bódai et al. (2015). Indeed, a codimension one manifold separates the basins of attraction of the W and SB states. We refer to $D^W$ ($D^{SB}$) as the basin of attraction of the W (SB state). We refer to $B$ as the basin boundary, which

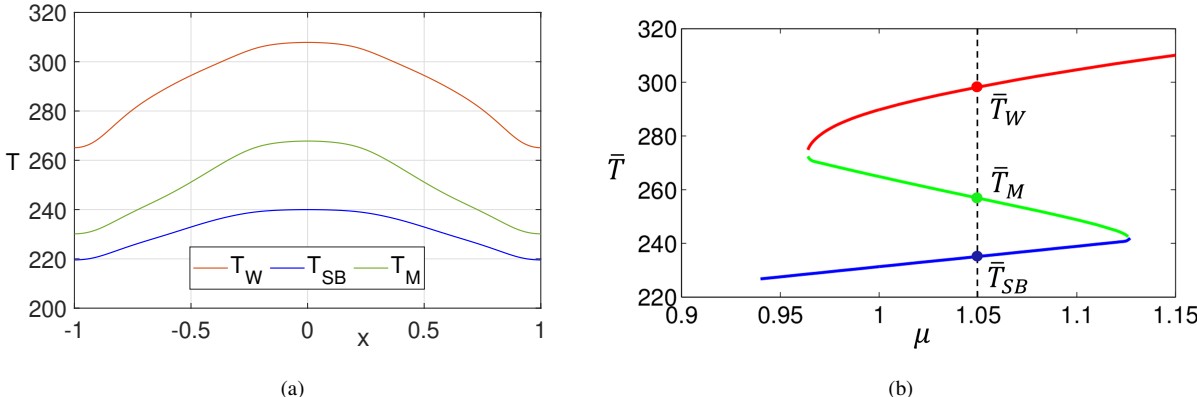

**Figure 1.** (a) Stationary solutions $T_W(x)$, $T_{SB}(x)$ and $T_M(x)$ in Kelvin (K) of zonally averaged energy-balance model (1). (b) Bifurcation diagram of the average temperature $\overline{T}$ as a function of control parameter $\mu$.

includes a single edge state $M$. Therefore, the system has three stationary solutions $T_W(x)$, $T_{SB}(x)$, and $T_M(x)$ for the W, SB, and M state, respectively, shown in Figure 1a. In Ghil (1976) the three stationary solutions were obtained by equating $T_t$ to 0,

and it was shown, through linear stability analysis, that the stationary solutions $T_W$ and $T_{SB}$ are stable, while $T_M$ is unstable. In Bódai et al. (2015) the unstable solution $T_M$ was constructed using a modified version of the edge tracking algorithm (Skufca et al., 2006).

Following previous studies (Bódai et al., 2015; Lucarini and Bódai, 2019; Lucarini and Bódai, 2020; Margazoglou et al., 2021), when visualising our results, we apply a coarse-graining to the phase space of the model. In what follows, we perform

a projection on the plane spanned by the spatially averaged temperature $\overline{T}$ and the averaged Equator minus Poles temperature difference $\Delta T$, defined as

$$\overline{T} = [T(x,t)]_0^1,$$ (10)

$$\Delta T = [T(x,t)]_0^{1/3} - [T(x,t)]_{1/3}^1, \text{ where}$$ (11)

$$[T(x,t)]_{x_l}^{x_h} = \frac{\int_{x_l}^{x_h} \cos(\pi x/2) T(x,t) dx}{\int_{x_l}^{x_h} \cos(\pi x/2) dx}.$$ (12)

Such a representation allows for a minimal yet still physically relevant description of the system. Indeed, changes in the energy budget of the system (warming versus cooling) are, to a first approximation, related to variations in $\overline{T}$, while the large-scale energy transport performed by the geophysical fluids is controlled by $\Delta T$. The boundary between high and low latitude in (11) is established at $x = \pm 1/3$, i.e. at $30°N/S$. Additionally, in some visualizations, we consider as a third coordinate the fraction of the surface with a below-freezing temperature (therefore we expect 1 for global glaciation and 0 for no ice). We refer to this

variable as $I$, and it is an attempt to extract an observable that resembles the sea-ice percentage of the Earth's surface. Thus, the stationary solutions $T_W(x)$, $T_{SB}(x)$, and $T_M(x)$, in terms of $\Delta T$ and $\overline{T}$, correspond to $\Delta T_W = 16$ K, $\Delta T_{SB} = 8.3$ K, $\Delta T_M = 17.5$ K; $\overline{T}_W = 297.7$ K, $\overline{T}_{SB} = 235.1$ K, $\overline{T}_M = 258$ K; and $I_W = 0.2$, $I_{SB} = 1$, and $I_M = 1$.

## 3 Background and Methodology

### 3.1 Stochastic Energy Balance Model.

In order to analyze the influence of random perturbations on the deterministic dynamics of the climate model described in Section 2, we perturb the relative intensity $\mu$ of the solar irradiance by including a symmetric $\alpha$-stable Lévy process and rewrite Eq. (1) in the form of the following stochastic partial differential equation (SPDE)

$$C(x)\mathcal{T}_t = D_I(x,\mathcal{T},\mathcal{T}_x,\mathcal{T}_{xx}) + D_{II}(x,\mathcal{T})(1 + \varepsilon/\mu \dot{L}^\alpha(t)) - D_{III}(\mathcal{T}), \tag{13}$$

where boundary and initial conditions defined by Eq. (2) apply to the stochastic temperature field $\mathcal{T}$. Here the parameter $\varepsilon > 0$ controls the noise intensity and $(L^\alpha(t)_{t \geq 0})$ is a symmetric $\alpha$-stable process defined in Appendix A. We consider symmetric processes because we want to have a simple mathematical model allowing for transitions in both the $SB \to W$ and the $W \to SB$ direction. Instead, a strongly skewed process would have made it very hard to explore the full phase space, because lack of symmetry would invariably favour one of the two transitions. As mentioned before, we refer to the Lévy case if the stability parameter $\alpha \in (0,2)$, so that we consider a jump process. We recall that the jumps become more frequent and less intense as $\alpha$ increases.

We define $\dot{\mathcal{L}}(t) = \mathcal{Q}(x)[1 - \alpha_a(x,\mathcal{T})]\dot{L}^\alpha(t)$, as the generalised derivative of a stochastic process in a suitably defined functional space. Equation (13) features multiplicative noise. The research interest on this type of SPDEs (Doering, 1987; Peszat and Zabczyk, 2007; Duan and Wang, 2014; Alharbi, 2021) is mainly focused on defining weak, strong, mild, and martingale solutions, and in specifying under which conditions these solutions exist and are unique, and in constructing numerical approximation schemes for the solutions (Davie and Gaines, 2000; Cialenco et al., 2012; Burrage and Lythe, 2014; Jentzen and Kloeden, 2009; Kloeden and Shott, 2001), among other aspects.

First let us define the concept of mild solution in this context. Let $(\Omega, \mathcal{F}, \mathbb{P})$ be a given complete probability space and $H(\|\cdot\|, \langle\cdot,\cdot\rangle)$ a separable Hilbert space with norm $\|\cdot\|$ and inner product $\langle\cdot,\cdot\rangle$. Equation (13) can be rewritten in the more general form as follows

$$\mathcal{T}_t = A(x)\left[E(x,\mathcal{T})\,\mathcal{T}_x\right]_x + F(x,\mathcal{T}) + \varepsilon G(x,\mathcal{T})\dot{L}^\alpha(t),$$
$$\mathcal{T}_x(-1,t) = \mathcal{T}_x(1,t) = 0, \tag{14}$$
$$\mathcal{T}(x,0) = T_0(x),$$

where $A, E, F, G$ are Lipschitz functions defined on $[-1,1] \times H$ and $G(x,\mathcal{T})\dot{L}^\alpha(t) = \dot{\mathcal{L}}(t)$. Under certain assumptions (Yagi, 2009), the problem (14) is formulated as a Cauchy problem whose local mild solution, a progressively measurable process $\mathcal{T}(t)$, for all $t \in [0,t_F]$ and $T_0 \in H$ has the following integral representation

$$\mathcal{T}(t) = \Psi(t)T_0 + \int_0^t \Psi(t-s)\Upsilon(\mathcal{T}(s))ds + \varepsilon \int_0^t \Psi(t-s)G(\mathcal{T}(s))d\beta + \varepsilon \int_0^t \Psi(t-s)G(\mathcal{T}(s))d\gamma, \tag{15}$$

where the dependence on $x$ is kept implicit and $\beta$ ($\gamma$) is the Poisson random measure (compensated Poisson random measure) defined through Lévy-Itô decomposition. Instead, $\Psi(t)$ with $t \geq 0$ is the evolution operator (Green's function in physical

terms) having the generalized semigroup property for the family of sector operators with the bounded inverses, and $\Upsilon(\mathcal{T}) = \mathcal{T} + F(x, \mathcal{T})$, $\mathcal{T} \in H$ is a nonlinear operator, which we assume to be Lipschitz continuous. Following the abstract theory presented in Yagi (2009), under certain structural assumptions for the operators $\Psi$ and $\Upsilon$ and for the functional space, one can prove that the solution (15) is the unique local mild solution of Eq. (14).

As mentioned above, things are radically different for the special case $\alpha = 2$, which corresponds to Gaussian noise. In this case, we revisit Eq. (14) and we define $\dot{L}^{\alpha=2}(t) = \dot{W}(t)$, where $(W(t)_{t \geq 0})$ is a Wiener process. We then define $\dot{\mathcal{W}}(t) = G(x, \mathcal{T})\dot{W}(t)$ as the generalised derivative of a Wiener process in a suitably defined functional space.

### 3.2 Noise-induced Transitions: Mean Escape Times

By incorporating stochastic forcing into the system, its long-time dynamics change significantly, allowing transitions between the competing basins. This dynamical behaviour is called metastability, and is graphically captured by Figure 2, where in plots (a-b) a typical spatio-temporal evolution of the temperature field is shown, for stability parameters $\alpha = 0.5$ and $\alpha = 1.5$, respectively. Instead, in plots (c-d) the temporal noise-induced evolution of global temperature $\overline{\mathcal{T}}$ and the averaged Equator and Poles temperature difference $\Delta \mathcal{T}$ (as defined in Eqs. (10)-(11)) is shown for the same $\alpha$. In what follows, we investigate the time statistics and the paths of the transitions between such basins.

In a complete probability space $(\Omega, \mathcal{F}, \mathbb{P})$ we define the first exit time $\tau_x$ of a cádlág mild solution $\mathcal{T}(\cdot; x)$ of (13) starting at $x \in D^{W/SB}$ domain of warm/snowball climate stable state as

$$\tau_x(\omega) = \inf\{t > 0 | \mathcal{T}_t(\omega, x) \notin D^{W/SB}\}, \quad \omega \in \Omega, \ x \in H. \tag{16}$$

The mean escape time is then expressed by $\mathbb{E}[\tau_x(\omega)]$. In the case of the infinite dimensional multistable reaction-diffusion system described by Chafee-Infante equation under the influence of additive infinite-dimensional $\alpha$-stable Lévy noise - $\alpha \in (0, 2)$ - it was shown (Debussche et al., 2013) that in the weak-noise limit $\varepsilon \to 0$ the mean escape time from one of the competing basins of attraction increases as $\varepsilon^{-\alpha}$. In such a limit the jump diffusion system reduces to a finite state Markov chain with values in the set of stable states. Details of this method are given in Appendix B. Similar results have been obtained for bistable one-dimensional SDEs (Imkeller and Pavlyukevich, 2006a, b). The basic reason behind this result is that, in order to study the transitions between the competing basins of attraction, one can treat the Lévy noise as a compound Poisson process where jumps arrive randomly according to a Poisson process and the size of the jumps $x$ is given by a stochastic process that obeys a specified probability distribution. For a symmetric $\alpha$-stable Lévy process, such a distribution asymptotically decreases as $|x|^{-1-\alpha}$, as discussed in Appendix A. Let's assume that positive values of $x$ bring the state of the system closer to the basin boundary (as in the case of positive fluctuations of the solar irradiance when studying escapes from the *SB* state). Assuming a simple geometry for the basin boundary, we have that a transition takes place when an event larger than a critical value $x_{crit} > 0$ is realised. The probability of such an event scales with $x_{crit}^{-\alpha}$. A similar argument applies when considering transitions triggered by negative fluctuations of the stochastic variable. Small-size events, which occur frequently and correspond to the non-occurrence of jumps, do not actually play any relevant role in determining the transitions, while they are responsible for the variability within each basin of attraction.

We now consider the case $\alpha = 2$. While the corresponding finite dimensional problem is thoroughly documented in the literature (Freidlin and Wentzell, 1984), and has been applied in a similar context by some of the authors (Lucarini and Bódai, 2019; Lucarini and Bódai, 2020; Ghil and Lucarini, 2020; Margazoglou et al., 2021), the treatment of infinite dimensional SDEs driven by an infinite dimensional Wiener process via the Freidlin-Wentzell theory requires further extension. In the present context, we refer to Budhiraja and Dupuis (2000) and Budhiraja et al. (2008) and references therein where the problem of an infinite dimensional reaction-diffusion equation driven by an infinite dimensional Wiener process has been addressed.

We assume that steady state conditions and ergodicity are met, and we also assume that the analysing system is bistable and a unique edge state is present at the basin boundary, as in the case studied here. In the case of Gaussian noise, transitions between the competing basins of attraction are not determined by a single event as in the $0 < \alpha < 2$ case, but, instead, occur as a result of very unlikely combinations of subsequent realisations of the stochastic variable acting as a forcing. In the weak-noise limit, the transitions occur according to the least unlikely (yet very unlikely) chain of events, whose probability is described using a large deviation law (Varadhan et al., 1985). One has that the mean escape time from either basin of attraction decreases exponentially with increasing noise intensity $\varepsilon$ and is given by a generalized Kramers' law

$$\mathbb{E}[\tau_{W/SB}(\varepsilon)] \approx \exp\left(\frac{2\Delta\Phi_{W\to M/SB\to M}(\mathcal{T})}{\varepsilon^2}\right), \tag{17}$$

where $\Delta\Phi_{W\to M} = \Phi_M(\mathcal{T}) - \Phi_W(\mathcal{T})$ is the height of the quasi-potential barrier in the W attractor, and, correspondingly $\Delta\Phi_{SB\to M}(\mathcal{T}) = \Phi_M(\mathcal{T}) - \Phi_{SB}(\mathcal{T})$ is the height of the quasi-potential barrier in the SB attractor, and $\Phi$ is the Graham's quasi-potential mentioned above (Graham, 1987; Graham et al., 1991).

### 3.3 Noise-induced Transitions: Most Probable Transition Paths

In the weak noise limit, the most probable path to escape an attractor is defined by a class of trajectories named "instantons" (Grafke et al., 2015; Bouchet et al., 2016; Grafke et al., 2017; Grafke and Vanden-Eijnden, 2019) or maximum likelihood escape paths (Lu and Duan, 2020; Dai et al., 2020; Hu and Duan, 2020; Zheng et al., 2020).. However, considering different noise laws result into possibly radically different instantonic trajectories (Dai et al., 2020; Zheng et al., 2020).

In our case, the theory indicates that if the stochastic forcing is Gaussian, under rather general hypothesis, the instanton will connect the attractor W/SB with the edge state M, which then acts as gateway for noise-induced transitions. Once the quasi-potential barrier is overcome, a free fall "relaxation" trajectory links M with the competing attractor SB/W. For equilibrium systems, (e.g. for gradient flows) where detailed balance is achieved, the relaxation and instantonic trajectories within the same basin of attraction are identical. On the contrary, for non-equilibrium systems, the relaxation and instantonic trajectories will differ, and will only meet at the attractor. See a detailed discussion of this aspect and of the dynamical interpretation of the quasi-potential $\Phi$ in Lucarini and Bódai (2020) and Margazoglou et al. (2021). Instead, if the noise is of Lévy type, the theory formulated for simpler equations suggests that the instanton will connect the attractor with a region on the basin boundary that is the nearest, in the phase space, to the attractor, as the concept of quasi-potential is immaterial (Imkeller and Pavlyukevich, 2006a, b).

In general, the maximum likelihood transition trajectory $\mathcal{T}_M(t)$ can be defined (Zheng et al., 2020; Lu and Duan, 2020) as
a set of system states at each time moment $t \in [0, t_f]$ that maximizes the conditional probability density function $p(\,.\,|\,.\,;\,.\,)$ of
passage from the origin stable state $\phi^{W/SB}$ to the destination stable state $\phi^{SB/W}$ and is expressed as

$$\mathcal{T}_M(t) = \arg\max_x \left[ p\left(\mathcal{T}(t) = x \,|\, \mathcal{T}(0) = x_0; \mathcal{T}(t_f) = x_f\right) \right] = \frac{p\left(\mathcal{T}(t_f) = x_f \,|\, \mathcal{T}(t) = x\right) \cdot p\left(\mathcal{T}(t) = x \,|\, \mathcal{T}(0) = x_0\right)}{p\left(\mathcal{T}(t_f) = x_f \,|\, \mathcal{T}(0) = x_0\right)}, \tag{18}$$

where $x_0$ ($x_f$) belongs to the basin of attraction $D^{W/SB}$ ($D^{SB/W}$) and $p(\,.\,|\,.\,)$ is the probability density function evolving
according to the Fokker-Planck equation (Risken, 1996). This method is applicable either if efficient numerical algorithms
are available to solve the Fokker-Planck equation associated to the studied stochastically driven system, or, empirically, when
considering a large ensemble of simulations. Note that this is not an asymptotic approach, i.e. it does not require a weak noise
limit $\varepsilon \to 0$ for its application and is applicable for systems with either Gaussian or non-Gaussian noise. Yet, in the weak-noise
limit, the definition (18) leads to constructing the optimal transition paths described above.

In the following section, for practical purposes, we construct such optimal transition path in the coarse grained 2D phase
space $(\overline{\mathcal{T}}, \Delta\mathcal{T})$ and 3D phase space $(\overline{\mathcal{T}}, \Delta\mathcal{T}, \mathcal{I})$ of the variables defined in Sect. 2 by averaging the ensemble of transitions
connecting the two competing states in the weak noise limit.

## 3.4 Numerical Methods

We solve Eq. (13) through the Matlab *pdepe* function, which is well suited for solving 1D parabolic and elliptic PDEs. We
discretize the 1D space with a regular grid of 201 gridpoints, following Bódai et al. (2015).

The time span of integration $t \in [0, T_f]$, varies for different cases, with $T_f \in (10^5, 15 \cdot 10^5)$ years, with time stepping of one
year. Each year, we consider a different value for the relative solar irradiance by extracting a random number $Z_j$, see Eq. (19).
To simulate the stochastic noise term $\varepsilon L^\alpha(t)$, which is added in the parameter $\mu$ in Eq. (13), we use the recursive algorithm
from Duan (2015). The process values $L^\alpha(t_1), ..., L^\alpha(t_N)$ at each moment $t_j$, $j \in \mathbb{N}$, are obtained via

$$L^\alpha(t_j) = L^\alpha(t_{j-1}) + (t_j - t_{j-1})^{\frac{1}{\alpha}} Z_j, \qquad j = 1, ..., N, \tag{19}$$

where the second term is an independent increment and $Z_j$ are the independent standard symmetric $\alpha$-stable random numbers
generated by an algorithm in Weron and Weron (1995). See also Grafke et al. (2015) for a detailed explanation of the steps
above. For illustrative reasons, some sample solutions of Eq. 13 for different values of $\alpha$ are shown in Figure 2 (a)-(b).

For the numerical simulations discussed below, we consider $\alpha = (0.5, 1.0, 1.5, 2)$ and $\varepsilon \in (0.0001, 0.3)$. We select $\varepsilon$ in such
a way that the noise intensity is strong enough to induce at least order of 10 transition, given our constraints in the time length
of the simulations, and weak enough that we are not far from the weak-noise limit, where the scaling laws discussed above
apply and transitions paths are well-organized. Our simulations are performed taking the Itô interpretation for the stochastic
equations.

We remark that when we consider Lévy noise, it does happen that for some years the solar irradiance has negative values. Of
course these conditions bear no physical relevance, and are a necessary result of considering unbounded noise. Nonetheless,
we have allowed for this to occur in our simulations in order to be able to stick to the desired mathematical framework. We

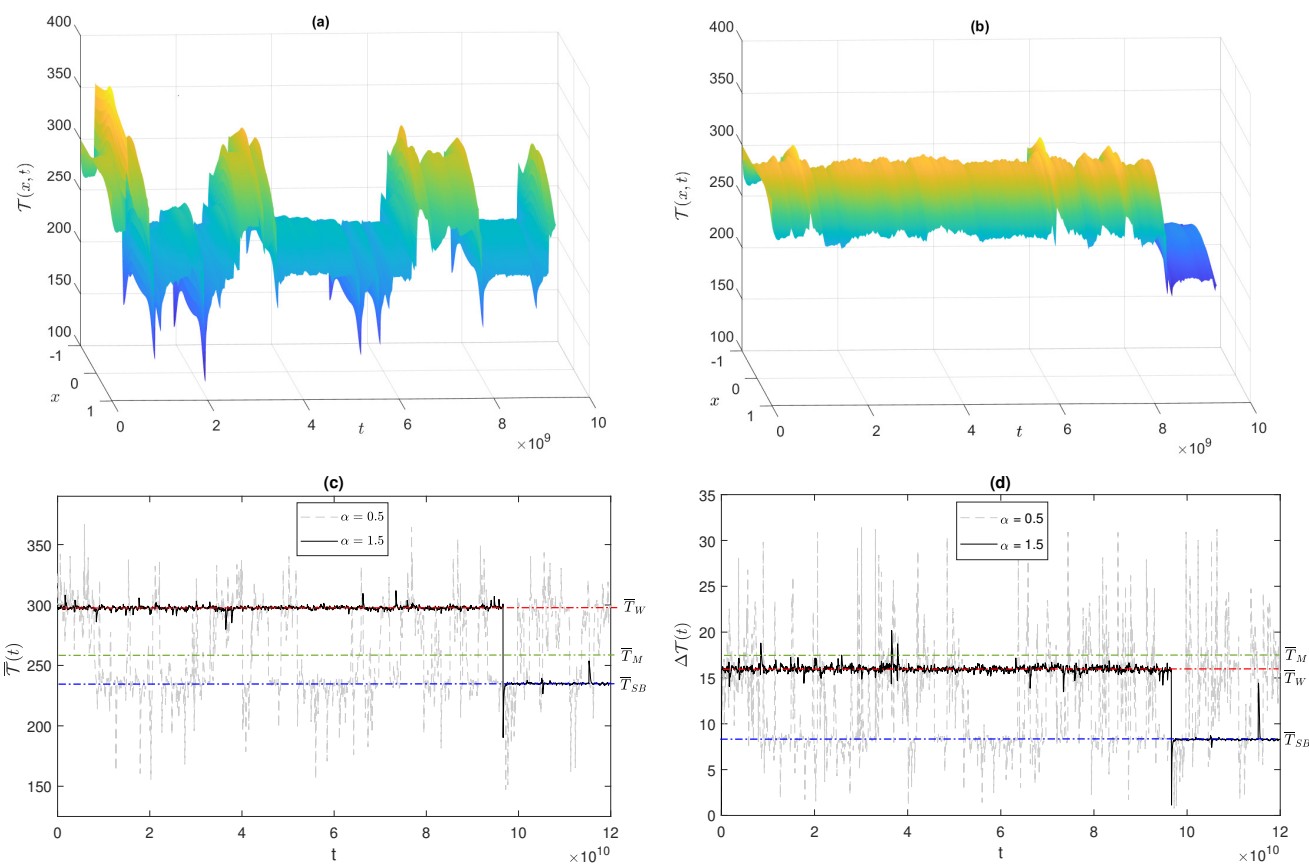

**Figure 2.** The metastable behavior of solution path of stochastic energy-balance model (13) for $\varepsilon = 0.04$, $T_0 = 300$ K, $t \in (0, 300)$ years (a) $\alpha = 0.5$, (b) $\alpha = 1.5$ and of (c) temperature average $\overline{\mathcal{T}}$ and (d) temperature contrast at low and high latitudes $\Delta\mathcal{T}$ for $\varepsilon = 0.01$, $\alpha = 0.5$, $\alpha = 1.5$. Red/green/blue dashed-dotted lines portray stationary climate states $\overline{T}_W/\overline{T}_M/\overline{T}_{SB}$, respectively.

remind that this study does not aim at capturing with any high degree of realism the description of the actual evolution of climate. At any rate, as can be understood from the discussion below in Sect. 4.2.2 and from what is reported in Appendix C, were we to consider longer lasting (e.g. 2 ys vs. 1 y) fluctuations of the solar irradiance, a satisfactory exploration of the transitions between the competing W and SB states would be possible with a greatly reduced occurrence of such unphysical events, the basic reason being the presence of a larger factor $(t_j - t_{j-1})^{\frac{1}{\alpha}}$ in Eq. 19 .

## 4 Results and Discussion

In what follows we aim at addressing three main questions: 1. What is the temporal statistics of the $SB \rightarrow W$ and $W \rightarrow SB$ transitions? 2. What are the typical transition pathways? 3. What are the fundamental differences between transitions caused by Gaussian vs. Lévy noise? A summary of the results of the numerical simulations is given in Table D1 in Appendix D,

**Table 1.** Estimates of exponent $\alpha$ via fitting of Eq. (B6) for the Lévy case (three first columns) and of energy barrier $\Delta\Phi_{W/SB\to M}$ via fitting of Eq. (17) for the Gaussian case (last column). In parenthesis is the estimated error of the last digit.

|  | | Lévy | | Gaussian |
| --- | --- | --- | --- | --- |
| $\alpha$ | 0.5 | 1.0 | 1.5 | 2 |
| W | 0.50(2) | 1.00(2) | 1.50(1) | $\Delta\Phi_{W\to M} = 0.068(1)$ |
| SB | 0.47(2) | 0.97(2) | 1.52(4) | $\Delta\Phi_{SB\to M} = 0.048(3)$ |

including sample size, i.e. number of transitions, point estimates for mean escape time and their 0.95-confidence intervals for exits from both the W and SB basins. See the Data Availability section for information on how to access the supplementary material (Lucarini et al., 2021) containing the raw data produced in this study as well as some illustrative animations portraying noise-induced transitions between the two competing metastable states.

## 4.1 Mean Escape Time

Our analysis confirms that there is fundamental dichotomy in the statistics of mean escape times between Lévy noise and Gaussian noise-induced transitions.

Figure 3a shows the dependence of the mean escape time from either attractor on $\varepsilon$ and $\alpha$ for the Lévy case. The red circles (blue squares) correspond to escapes from the W (SB) basin; see Lucarini et al. (2021) for additional details. The scaling $\propto \varepsilon^{-\alpha}$ presented in Eq. (B6) is shown by the dotted black line for each value of $\alpha$. We also portray the best power law fit of the mean residence time with respect to $\varepsilon$ for each value of $\alpha$; the confidence intervals of the exponent is shown in Table 1. Our empirical results seem to indicate, at least in this case, an agreement with the $\varepsilon^{-\alpha}$ scaling presented and discussed earlier in the paper. This points at the possibility that the $\varepsilon^{-\alpha}$ scaling might apply in more general conditions than what has been as of yet rigorously proven, and specifically when multiplicative Lévy noise is considered. The stochastically perturbed trajectories forced by Lévy noise consist of jumps, and the probability of occurrence of a high jump, which can trigger the escape from the reference basin of attraction, is polynomially small in noise intensity $\varepsilon$.

The Gaussian case - where no jumps are present - is portrayed in Fig. 3b. We show in semi-logarithmic scale the mean residence time versus $1/\varepsilon^2$. We perform a successful linear fit of the logarithm of the mean residence time in either attractor versus $1/\varepsilon^2$, and using Eq. (17), we obtain an estimate of the local quasi-potential barrier $\Delta\Phi_{W/SB\to M}$, which is half of the slope of the corresponding straight lines of the linear fit; see the last column of Table 1. We conclude that for $\mu = 1.05$ the local minimum of $\Phi$ corresponding to the W state is deeper than the one corresponding to the SB state.

## 4.2 Escape Paths for the Noise-Induced Transitions

We now explore the geometry of the transition paths associated with the metastable behaviour of the system. We first discuss the case of Gaussian noise because it is indeed more familiar and more extensively studied.

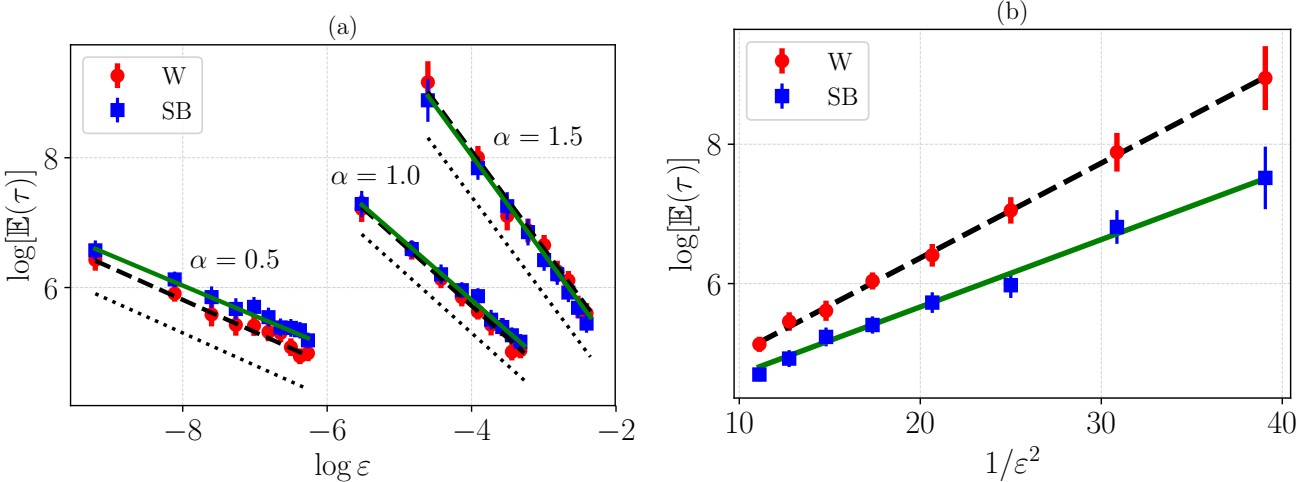

**Figure 3.** Estimates of the mean escape time $\mathbb{E}(\tau)$ (in y) from the W (blue circles) and SB (red squares) states as a function of the noise intensity $\varepsilon$. (a) Lévy noise for $\alpha = 0.5$, 1.0, 1.5, with the dotted line being the corresponding prediction from Eq. (B6), while the straight green (SB) and dashed black (W) are the fittings of Eq. (B6) of the relevant dataset. (b) Gaussian noise, with straight green (SB) and dashed black (W) being the fit of Eq. (17).

### 4.2.1 Gaussian noise

We estimate the transition paths by averaging among the escape plus relaxation trajectories using the run performed with the weakest noise, see Table D1. We first perform our analysis in the 2D-projected state space defined by $(\overline{\mathcal{T}}, \Delta\mathcal{T})$. We prescribe two small circular-shaped regions enclosing the two deterministic attractors and search the timeseries of the portions of the whole trajectory that leave one of such regions and reach the other one. This creates two subsets of our full dataset, from which we build a 2D histogram for each of the SB→W and W→SB transitions in the projected space. We then estimate the most

probable transition paths by finding for each bin value of $\overline{\mathcal{T}}$ the peak of histogram in the $\Delta\mathcal{T}$ direction. The distributions are very peaked, and almost indistinguishable estimates for the instantonic and relaxation trajectories are obtained when computing the average of $\Delta\mathcal{T}$ according to the 2D histogram conditional on the value of $\overline{\mathcal{T}}$.

In the background of Fig. 4 (a) we show the empirical estimate of the invariant measure in the 2D-projected state space defined by $(\overline{\mathcal{T}}, \Delta\mathcal{T})$. Additionally, we indicate the position of the deterministic attractors, where the blue (red) circle corresponds

to the SB (W) state, as well as of the M state (green square). In the inset of Fig. 4 (a) we present the ensemble of W→SB (SB→W) transitions as deep blue (red) contours. The most probable transition paths are shown in blue for the W→SB and in red for the SB→W. The instantonic portion of the blue (red) line is the one connecting the W (SB) attractor to the M state and is portrayed as a solid line, while the relaxation portion, connecting the M state with the SB (W) attractor, is portrayed as a dashed line. Within each basin of attraction, the instantonic and relaxation trajectories do not coincide, and, instead, only meet

at the corresponding attractor and at the M state. This is particularly clear for the W state. The presence of such a loop, proving

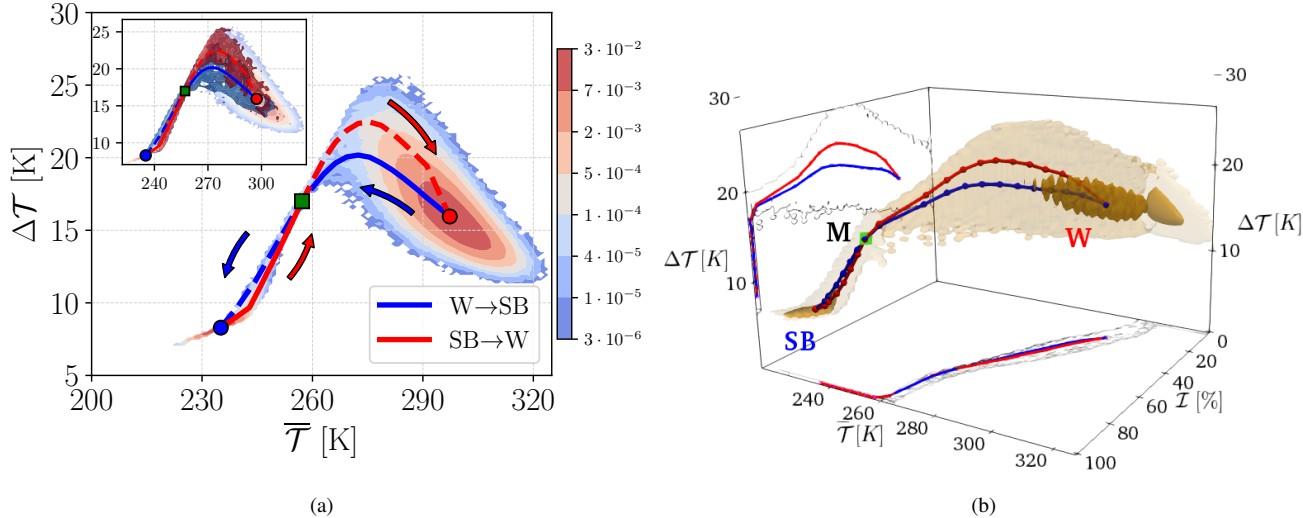

**Figure 4.** (a) Invariant measure in the 2D-projected state space defined by $(\overline{\mathcal{T}}, \Delta \mathcal{T})$. The colored points indicate the deterministic attractors of the SB(blue) M(green) and W(red) states and the blue (red) line is the stochastically averaged transition paths for the $W \rightarrow SB$ ($SB \rightarrow W$) transitions. Dashed (solid) lines are the relaxation (instantonic) trajectories. The arrows show the direction of transitions. Top left inset: The dark blue (red) contours portray the ensembles of the transition paths between $W \rightarrow SB$ ($SB \rightarrow W$). Here the system is driven by Gaussian noise with $\varepsilon = 0.14$. (b) Invariant measure and most probable transition paths ($W \rightarrow SB$ in blue and $SB \rightarrow W$ in red) in the 3D-projected state space defined by $(\overline{\mathcal{T}}, \Delta \mathcal{T}, \mathcal{I})$. The darker brown shading indicates higher probability density for the corresponding isosurface. 2D projections on the $(\overline{\mathcal{T}}, \mathcal{I})$ and $(\Delta \mathcal{T}, \mathcal{I})$ planes are shown. The location of the M state is given by a pink square. Here the system is driven by Gaussian noise with $\varepsilon = 0.16$.

the existence of non-vanishing probability currents and the breakdown of detailed balance, is a signature of non-equilibrium dynamics, which was also observed in Margazoglou et al. (2021) and has, instead, gone undetected in Lucarini and Bódai (2019); Lucarini and Bódai (2020). See the supplementary material for some illustrative simulations of the transitions.

Let's provide some physical interpretation of how the transitions occur. Looking at the $SB \rightarrow W$ most probable path, the escape includes a simultaneous increase in $\overline{\mathcal{T}}$ and $\Delta \mathcal{T}$. In practice, a $SB \rightarrow W$ transition takes place when, starting at the SB state, one has a (rare) sequence of positive anomalies in the fluctuating solar irradiance $\widetilde{\mu}$, i.e. $\widetilde{\mu} > \mu$. While the planet is warming globally, the Equator is warming faster than the Poles, resulting in a positive rate $\dot{\Delta \mathcal{T}} > 0$, because it receives, in relative and absolute terms, more incoming solar radiation. Considering that the Equator also in the SB state is warmer than the Poles, the melting of the ice conducive to the transition occurs first at the Equator, with a subsequent decrease of the albedo in this latitude. Once the system crosses the M state, and supposing that persistent $\widetilde{\mu} < \mu$ do not appear at this stage, the system will relax towards the W state. The relaxation includes a consistent global warming of the planet, but with a change of sign in the rate of $\dot{\Delta \mathcal{T}}$, and a subsequent decrease of $\Delta \mathcal{T}$ implying that as soon as the temperature at the Equator has risen enough, the Poles will then warm at a faster pace, because the ice-albedo effect kicks in.

The global freezing of the planet associated with the $W \to SB$ transition is qualitatively similar but not identical to the reverse $SB \to W$ process. Notice a considerable overlap of the transition paths ensembles in both basins of attraction, shown as red and blue contours in the inset of Fig. 4 (a). This implies the presence of less extreme non-equilibrium conditions compared to what observed in Margazoglou et al. (2021), where the $W \to SB$ and $SB \to W$ transitions occurred through fundamentally different paths; see discussion therein, especially regarding the role of the hydrological cycle.

Figure 4 (b) presents the optimal transition paths $W \to SB$ and $SB \to W$ in a three-dimensional projection where we add as third coordinate the variable $\mathcal{I}$, which indicates the fraction of the surface that has subfreezing temperatures ($\mathcal{T} < 273.15$ K). On the sides of the figure, two two dimensional projections on the $(\mathcal{T}, \Delta\mathcal{I})$ and on the $(\Delta\mathcal{T}, \mathcal{I})$ planes are shown. Here, darker brown shadings indicate higher density of points and the red and blue dots sample the highest probability for the $SB \to W$ and $W \to SB$ transitions paths, respectively. One could argue that the presence of an intersection between the $SB \to W$ and $W \to SB$ highest probability transition paths in Fig. 4 (a) could have been a simple effect of 2D projection. Instead, we see here that the $SB \to W$ and $W \to SB$ most probable transition paths also cross in the 3D projection in a well-defined region, which indeed corresponds to the M state (pink square).

#### 4.2.2 Lévy noise

There is scarcity of rigorous mathematical results regarding the weak-noise limit of the transition paths between competing states in metastable stochastic systems forced by multiplicative Lévy noise. Indeed, the derivation of analytical results for this type of systems largely remains an open problem. Recently, for stochastic partial differential equations with additive Lévy and Gaussian noise, the Onsager-Machlup action functional has been derived in Hu and Duan (2020), leading to a precise formulation of the most probable transition paths. Hence, we do not have solid mathematical results to interpret what we describe below, where, instead, we need to use heuristic arguments. As far as we know, this is the first attempt to estimate the most probable transition pathway between the metastable states in infinite stochastic systems with multiplicative pure Lévy process.

A striking feature in Figure 5 is that the invariant measure and the structure of the most probable transition paths (SB→W and W→SB), in the weak-noise limit, are fundamentally different between the Lévy case and the Gaussian one. The invariant measure is highly peaked (dark red in the color scheme) in a small region around the deterministic attractors, as most typically the Lévy noise fluctuations of $\widetilde{\mu}$ are very small. Additionally, the most probable transition paths depend very weakly on the chosen value for the stability parameter $\alpha$. This suggests that the geometry of most probable path of transitions does not depend on the frequency and height of the Lévy diffusion jumps, but rather on the qualitative fact that we are considering a jump process. Note that each panel of Fig. 5 is computed using data coming from the weakest noise considered for the corresponding value of $\alpha$, see Table D1.

The $W \to SB$ most probable transition path is characterized by the simultaneous decrease of both $\overline{\mathcal{T}}$ and $\Delta\mathcal{T}$. This implies that the jump leads to a rapid and direct freezing of the whole planet. The stochastically averaged path crosses the basin boundary far from the M state. The most probable $SB \to W$ transition follows, instead, a path that is somewhat similar to the one found for the Gaussian case. We then argue that the closest region in the basin boundary to SB attractor is not too far from the M

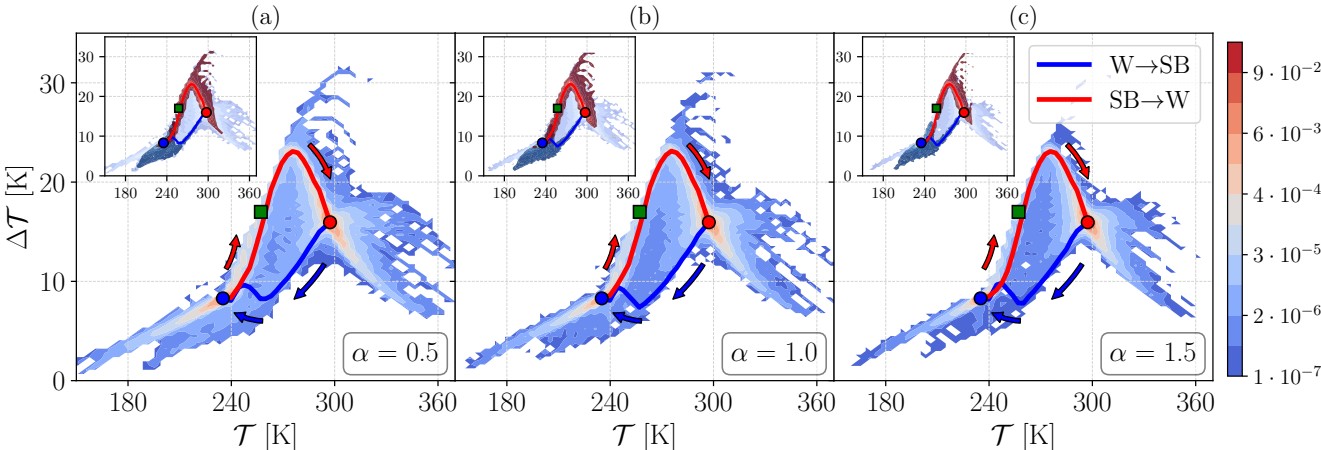

**Figure 5.** Two-dimensional projection of the invariant measure on $(\overline{\mathcal{T}}, \Delta\mathcal{T})$ for different choices of $\alpha$ for Lévy noise; (a) $\alpha = 0.5$ and $\varepsilon = 0.0001$, (b) $\alpha = 1$ and $\varepsilon = 0.004$, (c) $\alpha = 1.5$ and $\varepsilon = 0.01$. The blue (red) line corresponds to the W→SB (SB→W) most probable transition path, and the arrows show the direction of transitions. The colored points indicate the deterministic attractors of the SB(blue) M(green) and W(red) states. In the inset left top corner plot of (a-c) the dark blue (red) contours are the ensembles of the transition paths between $W \rightarrow SB$ ($SB \rightarrow W$).

state. Further insight on difference between the Gaussian and Lévy case can be found by looking at the animations included in the supplementary material.

### 4.2.3 Lévy noise - Singular Perturbations

Based on what is discussed in Sec. 3.2, we expect that the transitions occur through the nearest region to the outgoing attractor in the basin boundary. We now try to clarify the properties of the most probable escape paths in the Lévy noise case by considering an additional set of simulations, taking inspiration from the edge tracking algorithm (Skufca et al., 2006). The idea is to exploit the fact that large jumps drive the transitions in the Lévy noise case. Starting from the deterministic *SB* state, we apply in Eq. 6 singular perturbations of the form $\mu \rightarrow \mu + \kappa\delta(t)$ and bracket the critical value $\kappa_{crit}^{SB \rightarrow W}$ leading in a transition to the *W* state as $\kappa_{crit}^{SB \rightarrow W} \in [\kappa_{crit}^{SB \rightarrow W,s}, \kappa_{crit}^{SB \rightarrow W,u}]$, where the simulation performed with the supercritical (subcritical) value of $\kappa = \kappa_{crit}^{SB \rightarrow W,u}$ ($\kappa = \kappa_{crit}^{SB \rightarrow W,s}$) asymptotically reaches the competing (comes back to the initial) steady state. We obtain $\kappa_{crit}^{SB \rightarrow W,s} \approx 1.149$ *y* and $\kappa_{crit}^{SB \rightarrow W,u} \approx 1.1492$ *y*. Starting from the *W* state, we follow a similar procedure and find $\kappa_{crit}^{W \rightarrow SB} \in [\kappa_{crit}^{W \rightarrow SB,u}, \kappa_{crit}^{W \rightarrow SB,s}]$. We obtain $\kappa_{crit}^{W \rightarrow SB,s} \approx -1.3458$ *y* and $\kappa_{crit}^{W \rightarrow SB,u} \approx -1.346$ *y*. In both cases, the value of $\kappa$ of the supercritical and subcritical paths differ by $\delta\kappa \approx 0.0002$ *y*.

The projections on the 2D phase space spanned by $(\mathcal{T}, \Delta\mathcal{T})$ of the supercritical and subcritical paths corresponding to the $SB \rightarrow W$ ($W \rightarrow SB$) transition are shown in Fig. 6a (Fig. 6b) using the thick and thin black dashed lines, respectively. The basin boundary is indicated by a cyan (magenta) line for the $SB \rightarrow W$ ($W \rightarrow SB$) transition. The steps to estimate the basin boundary are presented in Appendix C. Note that we are using as background the invariant measure and the subset of transitions

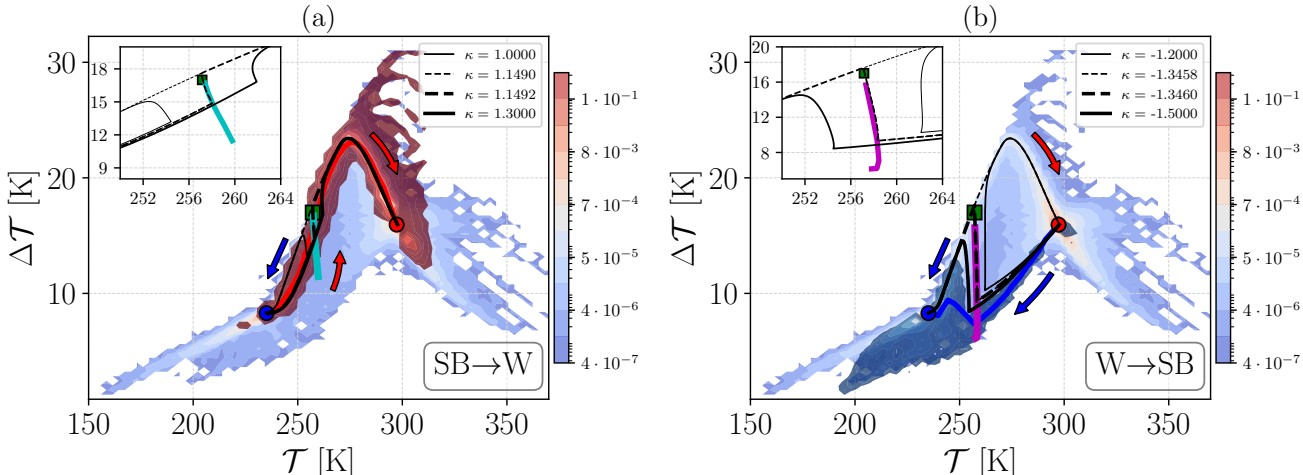

**Figure 6.** Comparison between supercritical and subcritical paths constricted using singular perturbations and the ensemble of trajectories corresponding to Lévy noise-induced transitions. Panel a): $SB \rightarrow W$ transitions. Panel b): $W \rightarrow SB$ transitions. Singular perturbations: the subcritical paths are depicted thin solid and dashed lines; the supercritical paths, leading to transition, are depicted as thick solid and dashed lines. See text for further details. Here $\alpha = 1.0$ and $\varepsilon = 0.004$.

referring to Lévy noise simulations performed using $\alpha = 1.0$. Nonetheless, what discussed below would apply equally well had we chosen to consider as background, instead, data coming from the simulation performed with $\alpha = 0.5$ or $\alpha = 1.5$. Indeed, for the link we propose between transitions due to Lévy noise and the case of singularly perturbed trajectories what matters depends on the discontinuous nature of the Lévy noise.

     In panel 6a (6b) the supercritical and subcritical paths are superimposed on the ensemble of trajectories of the $SB \rightarrow W$

($W \rightarrow SB$) transitions due to Lévy noise. The lines are better visible in the insets. By construction, after the perturbation is applied, the supercritical and subcritical orbits are close to the basin boundary. Hence, they are attracted towards the M state before being repelled towards the final asymptotic state. For comparison we also portray, for both the $SB \rightarrow W$ and the $W \rightarrow SB$ case, an additional pair of supercritical and subcritical paths that are constructed using values of $\kappa$ that differ by $\delta\kappa = 0.3$ $y$, which is a much larger difference than the one mentioned previously (for the dashed lines). The paths depicted as thick (thin)

solid lines cross (do not cross) the basin boundary. When looking at the $W \rightarrow SB$ transitions due to Lévy noise, we understand that if the perturbation sends the orbit near the basin boundary, the subsequent evolution of the system follows the supercritical paths. Of course, the Lévy perturbation often overshoots the basin boundary: in this case, after the transition, the orbit is not necessarily attracted towards the M state, whereas it converges directly to the final $SB$ state. The signature of the attracting influence of the M state persists in the stochastically averaged transition trajectory: note the bending towards higher values of

$\Delta\mathcal{T}$ before the eventual convergence to the $SB$ state. In the case of the $SB \rightarrow W$ transitions a similarly good correspondence between the supercritical path and the stochastically averaged transition trajectory is found.

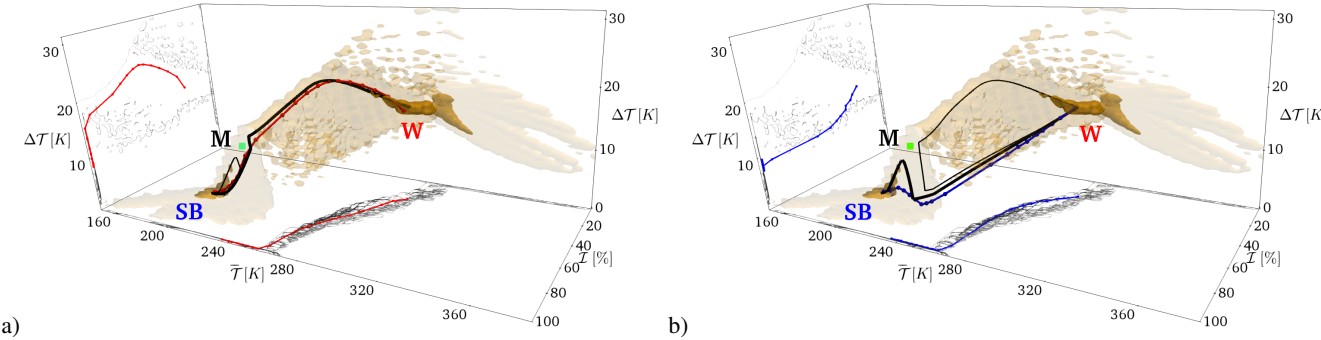

**Figure 7.** Same as panels a) and b) of Fig. 6, respectively, using the same 3D projection, lines, color coding as in Fig. 4. The subcritical and supercritical paths are depicted as thin and thick solid lines, respectively. The corresponding dashed lines shown in Fig. 6 are not reported here for simplicity. Here $\alpha = 1.0$ and $\varepsilon = 0.004$.

Further support to the viewpoint proposed here is given by Fig. 7a and Fig. 7b, which are constructed along the lines of Fig. 4b and portray the supercritical and subcritical paths as well as the stochastically averaged transition trajectory realised via Lévy noise with $\alpha = 1.0$ for the $SB \rightarrow W$ and $W \rightarrow SB$ case, respectively.

The transitions shown in Figs. 6-7 have been obtained by considering a discrete approximation of the Dirac's $\delta$ where the forcing acts at constant value for 1 $y$. Specifically, the Dirac's $\delta(t)$ is approximated as $\Delta_\tau(t)$, where $\Delta_\tau(t) = 1/\tau$ if $0 < t < \tau$ and vanishes elsewhere. The results are virtually unchanged if one considers $\tau < 1$ $y$ because the main dynamical processes of re-equilibration of the system act on longer time scales. The effect of the negative feedbacks of the system start to become apparent when considering slower perturbations, lasting 2 or more years. Indeed, the resilience of the system to transitions is reduced when, *ceteris paribus*, faster perturbations are considered; see analyses in this direction dealing with stability of the large scale ocean circulation (Stocker and Schmittner, 1997; Lucarini et al., 2005, 2007; Alkhayuon et al., 2019). Nonetheless, also in this case, the agreement with the results presented in Figs. 6-7 is considerable. We remark that considering longer lasting perturbations allows one to observe $W \rightarrow SB$ transitions without using at any time (unphysical) negative values for the solar irradiance. This is reassuring in terms of robustness and of physical sense of our results. Further details on the impact of the impact of considering different values of $\tau$ are reported in Appendix C.

## 5 Conclusions

It is a well-known that, as a result of the competition between the Boltzmann stabilizing feedback and the ice-albedo desta-bilizing feedback, under current astronomical and astrophysical conditions the climate system is multistable, as at least two competing and distinct climates are present, the W and the SB. More recent investigations indicate that the partition of the phase space of the climate system might be more complex, as more than two asymptotic states might be present, some of them, possibly, associated with small basins of attraction.

For deterministic multistable systems the asymptotic state of an orbit depends uniquely on the initial condition, and, specif-ically, on which basin of attraction it belongs to. The presence of stochastic forcing allows for transitions to occur between

competing basins, thus giving rise to the phenomenon of metastability. Gaussian noise as a source of stochastic perturbations has been widely studied by the scientific community in recent years and provided very fruitful insight of the multiscale nature of the climatic time series. However, it has become apparent that more general classes of $\alpha$-stable Lévy noise laws might also be suitable for modeling the observed climatic phenomena. In this regard, it is important to achieve a deeper understanding of the possible noise-induced transitions between competing stable climate states under $\alpha$-stable Lévy perturbations and compare them with the Gaussian case.

As a starting point in this direction, we have studied the influence of different noise laws on the metastability properties of the randomly forced Ghil-Sellers EBM, which is governed by a nonlinear, parabolic, reaction-diffusion PDE. In the deterministic version of the model, we have three steady-state solutions: two stable, attractive climate states and one unstable saddle, corresponding to the edge state. The stable states correspond to the well-known W and SB climates. There is a fundamental dichotomy in the properties of the noise-induced transitions determined by whether we consider a stochastic forcing of intensity $\varepsilon$ with a Gaussian versus an $\alpha$-stable Lévy noise law. Note that, instead, the spatial structure of the noise is unchanged. This indicates that the phenomenology associated with the metastable behaviour depends critically on the choice of the noise law. Not many studies have investigated, numerically or through mathematical theory, the properties of transitions in metastable systems driven by multiplicative Lévy noise, as done here.

First, in the weak noise limit $\varepsilon \to 0$, the mean residence times inside either competing basin of attraction for diffusions driven by Gaussian vs. Lévy noise have a fundamentally different dependence on $\varepsilon$. Our results show that the logarithm of the mean residence time for Gaussian diffusions scales with $\varepsilon^{-2}$, while, instead, a much weaker dependence is found for the Lévy case. Indeed, we find that the mean residence time is proportional to $\varepsilon^{-\alpha}$, where $\alpha$ is the stability parameter of the noise law. This result is in agreement with what has been proven in some special cases for additive Lévy noise, and might indicate that these scaling properties are more general than usually assumed. We propose a simple argument based on approximating the Lévy noise as a composed Poisson process to support the applicability of the result in general circumstances, but, clearly, detailed mathematical investigations in this direction are needed.

Secondly, the results obtained for the most probable transition paths confirm that, in the weak-noise limit, escapes from basins of attraction driven by Gaussian noise take place through the edge state. Additionally, instantonic and relaxation portions within each basin of attraction are clearly distinct, indicating nonequilibrium conditions, yet qualitatively similar. In turn, Lévy diffusions leave the basin through the boundaries region closest to the outgoing attractor, which seems to be the vicinity of the edge state when the thawing transition is considered. The freezing transition, instead, proceeds along a path that is fundamentally different. Finally, the most probable transition paths for the Lévy case appear to depend very weakly on the value of the stability parameter $\alpha$, but seem, instead, determined by the nature of the Lévy noise of being a jump process. Indeed, we suggest that these properties can be better understood by considering that, to a first approximation, the transitions due to Lévy diffusion correspond to supercritical paths associated with Dirac's delta-like singular perturbations to the solar irradiance. This viewpoint seems of general relevance in other problems where Lévy noise is responsible for exciting transitions between competing metastable states.

Our findings provide strong evidence that choosing noise laws other than Gaussian leads to fundamental changes in the metastability properties of a system, both in terms of statistics of the transitions between competing basins of attraction and most probable paths for such transitions. Leaving the door open for general noise laws might be relevant both for interpreting observational data and for performing modelling exercises for the climate system and complex systems in general.

Let's give an example of the impact of making a wrong assumption on the nature of the acting stochastic forcing. Were we to naively interpret one of the panels of Fig. 5 as resulting from the dynamics of a dynamical system perturbed by Gaussian noise, we would have to conclude that the unperturbed deterministic system possesses at least two edge states on the basin boundary separating the competing basins of attraction; see Margazoglou et al. (2021) for a case where this situation applies. Hence, we would infer fundamentally wrong properties on the geometry of the phase space. Additionally, we would infer fundamentally different properties for the drift term.

Recent developments in data-driven methods based on the formalism of the Kramers–Moyal equation allow to test accurately whether data are compatible with the hypothesis that stochasticity in the dynamics enters as a result of Gaussian noise or more general form of random forcing (Rydin Gorjão et al., 2021b; Li and Duan, 2022). Indeed, we point the reader to the recent contribution by Rydin Gorjão et al. (2021a), which shows that the analysis of proxy climatic datasets indicates the need to go beyond Langevin equation-based modelling, as they discover by that it is necessary to treat noise as the sum of continuous and discontinuous processes. This indicates the need to consider in future modelling exercises the possibility of investigating the properties of metastable systems where the stochastic forcing comes as the result of simultaneous Gaussian and $\alpha-$stable Lévy noise perturbations.

*Data availability.* All the data used to produce the figures contained in this paper are publicly available as supplementary material in Lucarini et al. (2021) through the data repository `https://doi.org/10.6084/m9.figshare.16802503`.

*Video supplement.* Illustrative animations portraying noise-induced transitions can be found in the supplementary material. We further uploaded the relevant material in the `YouTube` platform with links: Lévy SB→W, Lévy W→SB, Gaussian SB→W, Gaussian W→SB.

*Author contributions.* All authors contributed equally to this work.

*Competing interests.* No competing interests are present

*Acknowledgements.* The authors wish to thank P. Ashwin, R. Börner, J. I. Díaz, J. Duan, M. Ghil, T. Grafke, S. Kalliadasis, A. Laio, X.-M. Li, and G. Pavliotis for useful exchanges on various topics covered in this paper. VL wishes to thank M. Allen for suggesting to look into

3D projections of the phase space when studying transitions paths. The authors acknowledge the support provided by the EU Horizon 2020 project TiPES (grant No. 820970). VL acknowledges the support provided by the EPSRC project EP/T018178/1. This paper is dedicated to K. Hasselmann.

## Appendix A: Stochastic perturbations of Lévy type.

In this section we revise the basic properties of a symmetric $\alpha$-stable Lévy process in a Hilbert space in which the solutions to SPDE (13) are defined. We also repeat some properties in $\mathbb{R}^n$ space, that are more familiar to a wide audience of readers. It is pertinent to refer to the distribution law of Lévy increments, its characteristic function, the Lévy-Itô decomposition and the Lévy jump measure for a deeper study of the metastable behavior of the stochastic climate system (13). Let $(\Omega, \mathcal{F}, \mathbb{P})$ be a given complete probability space and $H(\|\cdot\|, \langle\cdot,\cdot\rangle)$ a separable Hilbert space with norm $\|\cdot\|$ and inner product $\langle\cdot,\cdot\rangle$. A stochastic process $(L^\alpha(t)_{t\geq 0})$ is a symmetric $\alpha$-stable Lévy process in $H$ if it satisfies:

1) $L^\alpha(0) = 0$, a.s..

2) Independent increments: for any $n \in \mathbb{N}$ and $0 \leqslant t_1 < t_2 < \cdots < t_{n-1} < t_n$ the vector

$$(L^\alpha(t_1) - L^\alpha(t_0), \ldots, L^\alpha(t_n) - L^\alpha(t_{n-1})) \tag{A1}$$

is a family of independent random vectors in $H$.

3) Stationary increments: for $0 \leqslant l < t$ random vectors $L^\alpha(t) - L^\alpha(l)$ and $L^\alpha(t-l)$ have the same law $\mathfrak{L}(.)$ in $H$

$$\mathfrak{L}(L^\alpha(t) - L^\alpha(l)) = \mathfrak{L}(L^\alpha(t-l)). \tag{A2}$$

This law in $\mathbb{R}^n$ is a symmetric $\alpha$-stable distribution $\mathfrak{L}(.) = S_\alpha((t-l)^{\frac{1}{\alpha}}, 0, 0)$, i.e., zero skewness and shift parameters, with a stability parameter $\alpha \in (0, 2]$ and a scaling parameter $(t-l)^{\frac{1}{\alpha}}$. The stable distribution by the generalised central limit theorem (Schertzer and Lovejoy (1997)) is a limit in distribution as $n \to \infty$ of the normalized sum $Y_n = \frac{1}{B_n} \sum_{i=1}^{n} (X_i - M_n)$ of n independent identically distributed random variables $X_i$, with a common probability distribution function $F(x)$, which does not necessarily have to have moments of both the first and second order. A necessary and sufficient condition for this is (Kuske and Keller (2000); Burnecki et al. (2015))

$$F(x) = [c_1 + r_1(x)] \, |x|^{-\alpha}, \qquad x < 0,$$
$$= 1 - [c_2 + r_2(x)] \, x^{-\alpha}, \qquad x > 0, \tag{A3}$$

with $0 < \alpha \leq 2$, $c_1$ and $c_2$ positive constants, $r_1(x) \to 0$ as $x \to -\infty$ and $r_2(x) \to 0$ as $x \to +\infty$. When this condition holds and $\alpha = 2$ we can set $B_n = h(n)$, where $h(n)$ satisfies $h^2 = n \ln h$, and the stable distribution is just the Gaussian law.

4) Sample paths are continuous in probability, i.e. for any $t \geqslant 0$ and $\eta > 0$

$$\lim_{l \to t} \mathbb{P}(\|L^\alpha(t) - L^\alpha(l)\| > \eta) = 0. \tag{A4}$$

For $\alpha \in (0,2)$ the symmetric $\alpha$-stable Lévy process in $\mathbb{R}^n$ has characteristic function of the form

$$\mathbb{E}\left[e^{i\langle u, L^{\alpha}(t)\rangle}\right] = e^{-C(\alpha)\, t\, \|u\|^{\alpha}}, \qquad u \in \mathbb{R}^n, \qquad t \geq 0, \tag{A5}$$

where $C(\alpha) = \pi^{-1/2} \frac{\Gamma((1+\alpha)/2)\Gamma(n/2)}{\Gamma((n+\alpha)/2)}$, and $\Gamma(.)$ is the Gamma function. In the case where $\alpha = 2$ we set $C(2) = 1/2$ and (A5) becomes the characteristic function of a standard Brownian motion. However, Brownian motion cannot be seen as a weak limit of $\alpha$-stable Lévy process because of the divergence $C(\alpha) \to \infty$ as $\alpha \to 2$. The properties of the sample paths of $L^{\alpha}(t)$ are, in fact, quite different for $\alpha = 2$ and $\alpha < 2$. Firstly, the $\alpha$-stable Lévy process is a discontinuous, pure jump process, while the Brownian motion has continuous paths. Secondly, the Brownian motion has moments of all orders, whereas $\mathbb{E}\,|L^{\alpha}(t)|^{\gamma} < \infty$ iff $\gamma < \alpha$. It can also be proved that the tails of $L^{\alpha}(t)$ are heavy, i.e. $\mathbb{P}\left(L^{\alpha}(t) > u\right) \sim u^{-\alpha}, \ u \to \infty$, quite the opposite of the exponentially light Gaussian tails. Moreover for $\alpha \in (0,1)$, the path variation of $L^{\alpha}(t)$ is bounded on finite time intervals, and unbounded for $\alpha \in [1,2)$.

Although neither the incremental nor the marginal distribution of a Lévy process in general are representable by the elementary functions, the Lévy motion is completely determined by the Lévy-Khintchine formula which specifies the characteristic function of the Lévy process.

If $L^{\alpha}(t)$ is a symmetric $\alpha$-stable Lévy process in $H$, then:

1) (Lévy-Khintchine formula) Its characteristic function is

$$\Lambda_t(h) = \mathbb{E}\left[e^{i\langle h, L^{\alpha}(t)\rangle}\right] = e^{t\,\psi(h)}, \qquad h \in H, \qquad t \geq 0,$$

where

$$\psi(h) = \int_H \left(e^{i\langle h,y\rangle} - 1 - i\langle h,y\rangle \mathbf{1}_{\{0 < \|y\| \leqslant 1\}}\right) \nu(dy), \tag{A6}$$

here $\mathbf{1}_S$ is the indicator function for a set $S$, taking 1 on $S$ and 0 otherwise, and $\nu$ is a Borel measure (so called the Lévy jump measure) in $H$ for which $\int_H (1 \wedge \|y\|^2)\nu(dy) < \infty$ with $1 \wedge \|y\|^2 = \min\{1, \|y\|^2\}$. A Borel measure, as well, can be defined as the expected value of the number of jumps of specified size $Q$ in the unit time interval, i.e. $\nu(Q) = \mathbb{E}N(1,Q)(\omega), \omega \in \Omega$.

2) (Lévy-Itô decomposition) For any sequence of positive radii $r_n \to 0$ and $\mathcal{O}_n = \{y \in H \mid r_{n+1} < \|y\| \leqslant r_n\}$ there exist a sequence of independent compensated compound Poisson processes $(\bar{L}_n(t))_{t \geqslant 0}, n \geqslant 0$ in $H$ with jump measures $\nu_n(B) = \nu(B \cap \mathcal{O}_n)$ for $B \in \mathcal{B}(H)$ the Borel $\sigma$-algebra in $H$ and $n \geqslant 1$, which satisfy $\mathbb{P}$-almost surely for all $t \geqslant 0$

$$L(t) = \sum_{n=1}^{\infty} \bar{L}_n + L_0(t), \tag{A7}$$

$$\bar{L}_n(t) = L_n(t) - t \int_H y \nu_n(dy), \quad n \geqslant 1. \tag{A8}$$

If $L^{\alpha}(t)$ is a symmetric $\alpha$-stable Lévy process in $\mathbb{R}^n$ with generating triplet $(0,0,\nu_{\alpha})$, then there exist an independent Poisson random measure $N$ on $\mathbb{R}^+ \times (\mathbb{R}^n \setminus \{0\})$ (quantifying the number of jumps of $L^{\alpha}(t)$) such that for eath $t \geqslant 0$,

$$L^{\alpha}(t) = \int_{\|y\| < 1} y \tilde{N}(t, dy) + \int_{\|y\| \geqslant 1} y N(t, dy), \tag{A9}$$

where $\tilde{N}(dt,dx) = N(dt,dx) - \nu_\alpha(dx)\,dt$ is the compensated Poisson random measure and $\nu_\alpha(dx)$ is the jump measure. The small $\|y\| < 1$ (large $\|y\| \geqslant 1$) jumps are controlled by $\tilde{N}(t,dy)$ ( $N(t,dy)$ ).

3) Its Lévy jump measure $\nu$ is symmetric in the sense that $\nu(-Q) = \nu(Q)$ for $Q \in \mathcal{B}(H)$ and has the specific geometry

$$\nu(Q) = \int_Q \nu(dy) = \int_Q \frac{dr}{r^{1+\alpha}} \sigma(ds), \tag{A10}$$

where $r = \|y\|$ and $s = y/\|y\|$ and $\sigma : \mathcal{B}(\partial B_1(0)) \to [0,\infty)$ is an arbitrary finite Radon measure on the unit sphere of $H$. The jump measure for a symmetric $\alpha$-stable Lévy motion $L^\alpha(t)$ in $\mathbb{R}^n$ is defined by

$$\nu_\alpha(du) = c(n,\alpha) \frac{du}{\|u\|^{n+\alpha}}, \tag{A11}$$

with the intensity constant $c(n,\alpha) = \frac{\alpha\Gamma((n+\alpha)/2)}{2^{1-\alpha}\pi^{n/2}\Gamma(1-\alpha/2)}$ where $\Gamma(.)$ is the Gamma function; see Duan (2015); Applebaum (2009).

One can come to a more intuitive interpretation of the stability parameter $\alpha \in (0,2)$ variation: for smaller values of $\alpha$, the process is characterized by higher jumps with a lower frequency. As $\alpha$ increases, jumps decrease in height and the frequency of their occurrence increases.

## Appendix B: Probabilistic theory for the Lévy noise-induced escape.

We briefly recapitulate here the main ideas behind the proof given in Debussche et al. (2013) of how the mean residence time in the competing metastable states of stochastically perturbed Chafee-Infante reaction-diffusion PDE scales with the intensity $\varepsilon$ of the additive $L(t)$ $\alpha-$stable Lévy noise that acts as stochastic forcing.

One proceeds by considering the decomposition of the driving Lévy process with regularly varying the jump measure $\nu$ into small $\xi^\varepsilon$ and large $\eta^\varepsilon$ jump components. Let $\Delta_t L = L(t) - L(t-)$ denote the jump increment of $L$ at time $t \geqslant 0$, and $\frac{1}{\varepsilon^\rho}$ for $\varepsilon, \rho \in (0,1)$ the jump height threshold of $L$. The process $\eta^\varepsilon$ is a compound Poisson process consisting of all jumps of height $\|\Delta_t L\| > \varepsilon^{-\rho}$ with intensity

$$\beta_\varepsilon = \nu\left(\frac{1}{\varepsilon^\rho} B_1^c(0)\right) \approx \varepsilon^{\alpha\rho}, \tag{B1}$$

and the jump probability measure outside the ball $\frac{1}{\varepsilon^\rho} B_1(0)$ by

$$\nu\left(\cdot \cap \frac{1}{\varepsilon^\rho} B_1^c(0)\right) / \beta_\varepsilon, \tag{B2}$$

where $B_1(0)$ is a ball of unit radius in $H$ centered at the origin. The occurrence time of a $k$-th large jump is defined recursively by

$$\mathcal{Z}_0 = 0, \quad \mathcal{Z}_k = \inf\{t > \mathcal{Z}_{k-1} \mid \|\Delta_t L\| > \varepsilon^{-\rho}\}, \quad k \geqslant 1. \tag{B3}$$

The waiting times between sucessive $\eta_t^\varepsilon$ jumps have an exponential distribution $\mathcal{Z}_k - \mathcal{Z}_{k-1} \sim \text{Exp}(\beta_\varepsilon)$.

Small jump processes $\xi^\varepsilon = L - \eta^\varepsilon$ due to the symmetry of Lévy measure $\nu$ is a mean zero martingale in $H$ with finite exponential moments. Probabilistic events causing small jumps in the stochastic solution of the system are not able to overcome the "force" of its deterministic stable state and therefore, do not contribute to the exit from the basin of attraction. Formally, during the time between two large jumps $t_k = \mathcal{Z}_k - \mathcal{Z}_{k-1}$, the solution of (13) following the deterministic path (1) returns to a small vicinity of the stable equilibria $\phi^{W/SB}$

$$\sup_{x \in D^{W/SB}} \sup_{\mathcal{Z}_{k-1} \leqslant t \leqslant \mathcal{Z}_k} \|\mathcal{T}(t) - T(t)\| \to 0 \qquad \text{for} \qquad \varepsilon \to 0. \tag{B4}$$

When a first large jump occurs, the solution process moves to the neighboring domain of attraction with probability

$$\mathbb{P}(\phi^{W/SB} + \varepsilon \Delta_{t_i} L \notin D^{W/SB}) = \mathbb{P}\left(\Delta_{t_i} L \in \frac{1}{\varepsilon}[(D^{W/SB})^c - \phi^{W/SB}]\right)$$

$$= \frac{\nu(\frac{1}{\varepsilon}[(D^{W/SB})^c - \phi^{W/SB}] \cap \frac{1}{\varepsilon^\rho} B_1^c(0))}{\nu(\frac{1}{\varepsilon^\rho} B_1^c(0))} \approx \varepsilon^{\alpha(1-\rho)}. \tag{B5}$$

This is the probability that at time $t_i$ there will be a jump increment $\Delta_{t_i} L$ that exceeds the distance between the attractor and its domain of attraction boundary, expressed by the jump probability measure (B2). In the zero-noise limit the mean residence time in a basin of attraction is given by

$$\mathbb{E}[\tau(\varepsilon)] \approx \sum_{i=1}^{\infty} \mathbb{E}[\mathcal{Z}_i] \, \mathbb{P}[\inf\{j : \phi^{W/SB} + \varepsilon \Delta_{t_j} L \notin D^{W/SB}\} = i]$$

$$\approx \mathbb{E}[t_1] \, \mathbb{P}(\phi^{W/SB} + \varepsilon \Delta_{t_1} L \notin D^{W/SB}) \cdot \sum_{i=1}^{\infty} i \, (1 - \mathbb{P}[\phi^{W/SB} + \varepsilon \Delta_{t_1} L \notin D^{W/SB}])^{i-1}$$

$$\approx \frac{1}{\varepsilon^{\alpha\rho}} \varepsilon^{\alpha(1-\rho)} \left(\frac{1}{\varepsilon^{\alpha(1-\rho)}}\right)^2 = \frac{1}{\varepsilon^\alpha}, \tag{B6}$$

i.e. by the sum of all the mean values of large jump occurrence time times the probability that the minimum of a sample of size $i$ of jump increments is sufficiently large to get into the neighboring domain of attraction. Thus, at the random time instant of large jumps, the solution process transitions, in an abrupt move, from one attractor to another. Such behavior of the random dynamical system is known as a metastability.

In Debussche et al. (2013) it was proved that in the time scale $\lambda(\varepsilon) = \nu(\frac{1}{\varepsilon} B_1^c(0))$, $\varepsilon > 0$ the metastable shifting of the diffusion process between neighborhoods of the two attractors represents a continuous time Markov chain in state space $\{\phi^{SB}, \phi^W\}$ with a transition rate matrix $\mathfrak{Q}$

$$\mathfrak{Q} = \frac{1}{\mu(B_1^c(0))} \begin{pmatrix} -\mu((D^{SB} - \phi^{SB})^c) & \mu((D^{SB} - \phi^{SB})^c) \\ \mu((D^W - \phi^W)^c) & -\mu((D^W - \phi^W)^c) \end{pmatrix}, \tag{B7}$$

where $\mu(\cdot)$ is the limit measure of $\nu$.

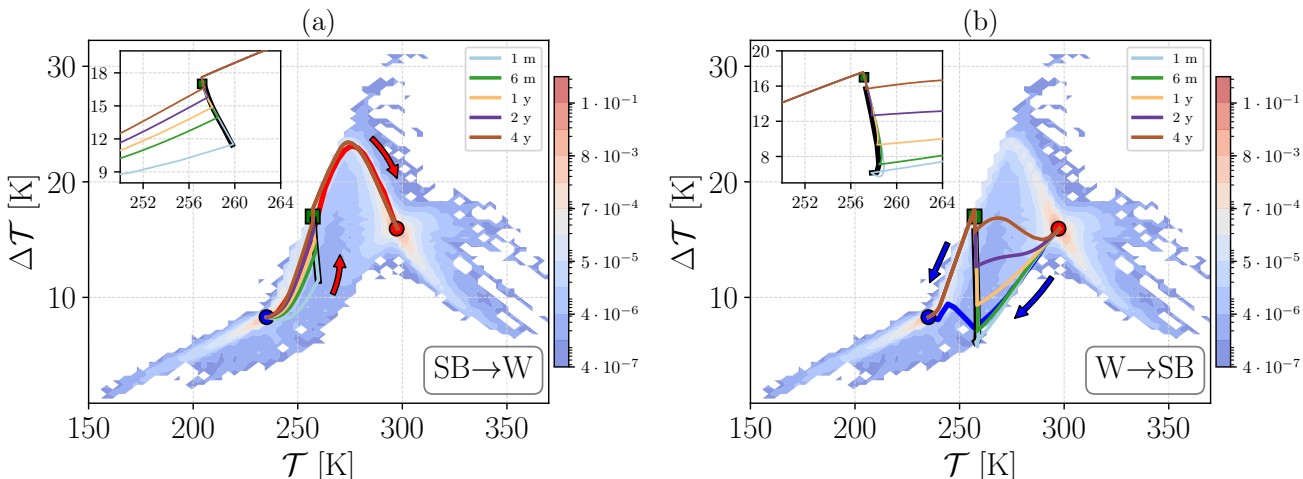

**Figure C1.** Comparison between supercritical paths constructed using singular perturbations of different duration $\tau$. Panel a): $SB \rightarrow W$ transitions. Panel b): $W \rightarrow SB$ transitions. Different values of $\tau$ correspond to different colored lines. The basin boundaries are now depicted as thick black lines. Here $\alpha = 1.0$ and $\varepsilon = 0.004$.

**Table C1.** Values of supercritical $\kappa_{crit}^{\cdots,u}$ for SB→W (second column) and W→SB (third column) for different approximations $\Delta_\tau(t)$ of the Dirac's $\delta(t)$.

| $\tau$ | $\kappa_{crit}^{SB \rightarrow W,u}$ (y) | $\kappa_{crit}^{W \rightarrow SB,u}$ (y) |
|---|---|---|
| 1 month | 1.105 | $-1.284$ |
| 6 months | 1.125 | $-1.313$ |
| 1 year | 1.1492 | $-1.346$ |
| 2 years | 1.199 | $-1.410$ |
| 4 years | 1.287 | $-1.542$ |

## Appendix C: Transitions induced by singular perturbations

In Sect. 4.2.3, and in particular Figs. 6-7 we have studied the effect of singular perturbations of a Lévy kick. The idea is that transitions in a system perturbed by Lévy noise are primarily driven by rare large jumps. By applying a singular perturbation of the form $\mu \rightarrow \mu + \kappa\delta(t)$ (where $\mu = \mu_0 = 1.05$ throughout) we have been able to bracket the critical values $\kappa_{crit}^{W \leftrightarrow SB,s}$ allowing for transitions between the two attractors. The expression $\kappa\delta(t)$ is approximated as $\kappa\Delta_\tau(t)$ where $\Delta_\tau(t) = 1/\tau$ if $0 < t < \tau$ and vanishes elsewhere. In Sect. 4.2.3 the results are shown for $\tau = 1y$.

We performed additional simulations to locate the supercritical and subcritical values of $\kappa$ for $\tau = 1$ month (m), 6 m, 2 y, and 4 y. The corresponding supercritical values of $\kappa_{crit}^{SB \rightarrow W,u}$ and $\kappa_{crit}^{W \rightarrow SB,u}$ are shown in Table C1. In Fig. C1 we plot the

corresponding supercritical transition trajectories for the values of Tab. C1 at different duration. Notice that now we use colored solid lines for the supercritical cases. To estimate the basin boundary we record the final point of when the forcing was active, in the $(\mathcal{T}, \Delta\mathcal{T})$ projected space, for each duration. This point for each case is particularly visible in the insets of Fig. C1, as a rapid reflection of the trajectory, which then follows closely the basin boundary (depicted as a thick black line). The basin boundary we can explore through this procedure is then estimated by linking the points obtained when considering various values of $\tau$. Notice that the estimated basin boundaries are slightly different when looking at the two SB→W and W→SB transitions, as the basin boundaries has folds than cannot be captured in the simpled 2-dimensional projection used in Fig. C1.

Finally, as stated earlier, from the third column of Table C1, we remark that, when considering forcings with duration of e.g. 2 y and longer, transitions from the W to the SB state can be achieved while retaining at all times a positive value for the solar irradiance, because while the forcing is active its value is $\mu + \kappa/\tau$.

## Appendix D: Estimates for the mean escape time

We report in Table D1 a summary of the statistics of the escape times from the W state and from the SB state for various choices of the noise law.

**(a)**

| $\varepsilon$ | 0.0001 | 0.0003 | 0.0005 | 0.0007 | 0.0009 | 0.0011 | 0.0013 | 0.0015 | 0.0017 | 0.0019 |
|---|---|---|---|---|---|---|---|---|---|---|
| Nº SB → W | 74 | 121 | 162 | 193 | 191 | 218 | 239 | 264 | 286 | 307 |
| Nº W → SB | 74 | 121 | 162 | 194 | 191 | 218 | 239 | 265 | 287 | 307 |
| $\mathbb{E}\left[\tau_{SB}\right]$ | 715 | 457 | 348 | 290 | 299 | 255 | 218 | 216 | 208 | 178 |
| $CI_{0.95}\left[\tau_{SB}\right]$ | [627,803] | [409,504] | [298,397] | [246,333] | [263,335] | [222,288] | [190,245] | [191,241] | [185,232] | [158,198] |
| $\mathbb{E}\left[\tau_{W}\right]$ | 618 | 367 | 265 | 226 | 224 | 203 | 200 | 160 | 139 | 146 |
| $CI_{0.95}\left[\tau_{W}\right]$ | [540,695] | [329,404] | [221,310] | [189,263] | [191,256] | [177,229] | [172,228] | [141,179] | [124,154] | [130,162] |

**(b)**

| $\varepsilon$ | 0.004 | 0.006 | 0.01 | 0.014 | 0.018 | 0.022 | 0.026 | 0.03 | 0.034 | 0.038 |
|---|---|---|---|---|---|---|---|---|---|---|
| Nº SB → W | 35 | 50 | 90 | 121 | 152 | 186 | 224 | 255 | 273 | 328 |
| Nº W → SB | 35 | 51 | 90 | 121 | 152 | 187 | 224 | 256 | 273 | 329 |
| $\mathbb{E}\left[\tau_{SB}\right]$ | 1461 | 1029 | 568 | 388 | 344 | 265 | 249 | 202 | 189 | 160 |
| $CI_{0.95}\left[\tau_{SB}\right]$ | [1235,1687] | [872,1186] | [482,654] | [336,441] | [306,382] | [230,301] | [216,281] | [178,227] | [166,211] | [143,176] |
| $\mathbb{E}\left[\tau_{W}\right]$ | 1357 | 925 | 531 | 431 | 313 | 270 | 197 | 187 | 177 | 144 |
| $CI_{0.95}\left[\tau_{W}\right]$ | [1124,1589] | [782,1067] | [461,600] | [382,481] | [279,347] | [232,307] | [171,223] | [164,211] | [156,197] | [127,161] |

**(c)**

| $\varepsilon$ | 0.01 | 0.015 | 0.025 | 0.035 | 0.045 | 0.055 | 0.065 | 0.075 | 0.085 | 0.095 |
|---|---|---|---|---|---|---|---|---|---|---|
| Nº SB → W | 5 | 8 | 21 | 37 | 57 | 78 | 118 | 142 | 170 | 231 |
| Nº W → SB | 5 | 9 | 22 | 37 | 58 | 79 | 118 | 142 | 170 | 231 |
| $\mathbb{E}\left[\tau_{SB}\right]$ | 7226 | 4410 | 2025 | 1383 | 800 | 629 | 418 | 308 | 282 | 191 |
| $CI_{0.95}\left[\tau_{SB}\right]$ | [5473,8979] | [3458,5362] | [1526,2525] | [1103,1664] | [677,923] | [529,729] | [366,470] | [256,361] | [242,323] | [165,217] |
| $\mathbb{E}\left[\tau_{W}\right]$ | 9544 | 6199 | 2418 | 1249 | 904 | 637 | 425 | 395 | 304 | 241 |
| $CI_{0.95}\left[\tau_{W}\right]$ | [7402,11686] | [4772,7625] | [1877,2959] | [1033,1464] | [770,1037] | [546,727] | [363,487] | [329,460] | [259,350] | [207,276] |

**(d)**

| $\varepsilon$ | 0.14 | 0.16 | 0.18 | 0.20 | 0.22 | 0.24 | 0.26 | 0.28 | 0.30 |
|---|---|---|---|---|---|---|---|---|---|
| Nº SB → W | 1 | 9 | 27 | 64 | 109 | 155 | 217 | 269 | 360 |
| Nº W → SB | 1 | 10 | 27 | 64 | 109 | 156 | 217 | 269 | 359 |
| $\mathbb{E}\left[\tau_{SB}\right]$ | 5038 | 1838 | 908 | 394 | 307 | 222 | 187 | 137 | 109 |
| $CI_{0.95}\left[\tau_{SB}\right]$ | [2378,7698] | [1138,2538] | [745,1071] | [328,460] | [268,345] | [198,246] | [164,210] | [120,154] | [99, 119] |
| $\mathbb{E}\left[\tau_{W}\right]$ | 25870 | 7700 | 2656 | 1153 | 605 | 418 | 273 | 234 | 168 |
| $CI_{0.95}\left[\tau_{W}\right]$ | [15339,36401] | [4991,10410] | [2121,3191] | [942,1364] | [524,686] | [377,459] | [238,308] | [208,259] | [151,186] |

**Table D1.** Estimates and 95%-confidence intervals for the mean escape time $\tau$ from the W state and from the SB state for Lévy noise with (a) $\alpha = 0.5$, (b) $\alpha = 1.0$, (c) $\alpha = 1.5$, and (d) Gaussian noise. Nº indicates the average number of transitions occurring in $10^5$ years of temporal evolution.

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
