# Peer review of "Lévy-noise versus Gaussian-noise-induced Transitions in the Ghil-Sellers Energy Balance Model"

_Nonlinear Processes in Geophysics, 2021_

## Referee Comment (RC2)

**A review of "Lévy-noise versus Gaussian-noise-induced Transitions in the Ghil-Sellers Energy Balance Model " by Lucarini et al., NPG-2021-34**

**General comment**

This manuscript studies the impact on the Ghil-Sellers energy balance model (EBM) of strongly non-Gaussian $\alpha$-stable Lévy fluctuations ($0 < \alpha < 2$) of solar irradiance, whereas mainly the case of Gaussian fluctuations ($\alpha$ =2) has been considered so far, including on a wide range of metastable systems. By means of numerical simulations for $\alpha = \{0.5; 1.0; 1.5\}$, the authors show the existence of an $\varepsilon^{-\alpha}$ scaling law of the mean residence time (in each metastable state) under a vanishing $\alpha$-stable Lévy multiplicative forcing intensity ($\varepsilon \rightarrow 0$, $\varepsilon|_{\alpha=0.5} \in [0.0001: 0.002]$; $\varepsilon|_{\alpha=1.0} \in [0.004: 0.04]$ and $\varepsilon|_{\alpha=1.5} \in [0.01: 0.1]$ numerically). Thus, the main and important result obtained is an extension to the Ghil-Sellers EBM of the scaling law found on simpler models (e.g., Imkeller and Pavlyukevich, 2006a,b; Debussche et al., 2013). This is certainly of great interest to NPG readership who have some familiarity with climate models and stochastic processes of various types.

However, in its very construction, the content of the manuscript itself seems excessively broad to allow readers to assess the generality/robustness and thus the impact of the results obtained. Several improvements seem to be needed before this article can be published. I hope the attached comments will be helpful for the revision.

**Detailed comments**

**Introduction of the Ghil-Sellers energy balance model**

The introduction of this model, with the help of Eqs 1-6 and its 9 parameters and 6 empirical functions (of location $x$), is quite abrupt whereas it is later summarised in three *physical* terms ( $D_I - D_{III}$), in particular in Eq. 10. It would be useful to proceed in the opposite direction and address rather usual questions such as the sensitivity of the model to the details of these terms, and in particular to the values of their parameters, as well as the uncertainty on the empirical estimates. These questions are particularly important regarding the scaling law obtained, e.g., its robustness.

**Introduction of the $\alpha$-stable Lévy noises**

The introduction of the $\alpha$-stable Lévy noises is a bit surprising and disappointing. Whereas the manuscript includes a considerable list of various applications of $\alpha$-stable Lévy noises, which can be understood as a vague argument of their potential interest for climate models, two main geophysical applications of the $\alpha$-stable Lévy noises were forgotten:

- The multiplicative cascades generated by $\alpha$-stable Lévy noises, often called "universal multifractals", have not only been widely used in geophysics but have been inspired by them (Schertzer and Lovejoy, 1987), specifically to analyse and simulate their ubiquitous intermittency and heavy tailed statistics, including at climate scales. This origin has been recognised by mathematicians who have used the term "Lévy multiplicative chaos" to emphasise their generality (e.g. Fan, 1987).
- The fractional Fokker-Planck equations for nonlinear SDE forced by non-Gaussian stable Lévy noises (e.g. Schertzer et al (2001) and references therein) used to analyse and simulate diffusion.

The authors argue for an $\alpha$-stable Lévy forcing by referring to the paleoclimatic records exhibiting strong non-Gaussian behaviour. It may be worth mentioning though that these observed heavytailed distributions generally do not support a power law exponent $\alpha < 2$, but a larger one that can be deduced from (i), see e.g. Schmitt et al (1995).

It is also surprising that the fundamental and common property of $\alpha$-stable Lévy ($0 < \alpha < 2$) and Gaussian ($\alpha = 2$) noises to be both stable (with a precise stability meaning of the index $\alpha$) and attractive (often presented like the generalised central limit theorem) is not presented. The fundamental statistical difference between $\alpha$-stable Lévy and Gaussian extremes is nor clearly presented, although indirectly evoked: namely $\alpha$-stable Lévy have heavy tails and $\alpha$ ($<2$) is then both the critical exponent of the divergence of statistical moments and the exponent of these power-law tails, whereas Gaussian noises do not have these extreme behaviours. Similarly, the scaling law of the increments (nonetheless used in Eq.16) and that of the Lévy jump measure are not mentioned (only indirectly in Eq. A7), whereas they play a key role. Moreover, only symmetric $\alpha$-stable Lévy noises are introduced, whereas unlike Gaussian noises, $\alpha$-stable Lévy noises are easily skewed and only extremely skewed $\alpha$-stable Lévy noises can generate multiplicative cascades (i) with finite (theoretical) statistics and avoid spurious (empirical) estimates. The latter may impact the rationale (L 309-312) to limit the range of intensity $\varepsilon$.

Overall, it seems that Appendix A is unbalanced mainly recalling properties of Lévy processes instead of the specific properties of the $\alpha$-stable Lévy processes. On the contrary, the physical properties of the latter recapitulated above may help to answer the question: is there a physical reason to escape the "Gaussian rigidity" and to consider $\alpha$-stable Lévy fluctuations?

**Robustness of the result and the multiplicative nature of the forcing term**

As already mentioned, it is necessary to discuss the robustness of the result, especially because of the many parameters and empirical functions involved in the climate model. Since the numerically observed scaling corresponds to that of an additive stochastic forcing, it is suggested to first assess the multiplicative character, i.e., whether the stochastic forcing in such a regime or only very weakly multiplicative. This depends exclusively on the effective temperature dependence of the albedo term (Eq. 5). It could be then important to estimate the latter.

**Negative values of the solar irradiance**

Despite the initially strong physical orientation of the manuscript, the authors do not hesitate to finally abandon some physical relevance, accepting negative values of the solar irradiance, "to be able to stick to the desired mathematical framework" (L 314-316), which is in turn substituted by a numerical framework with limited ranges of noise intensity $\varepsilon$. These can contribute to overlooking the fact that symmetric $\alpha$-stable Lévy noises generate much more frequent negative fluctuations than Gaussian noises and thus a significant gap between mathematical convenience and physics. The authors partly acknowledge this problem in their conclusions in terms of general heuristics (e.g., L 449-452), but they could be more precise in their critical analysis.

**Minor comments**

- L 124: the expression "discontinuous càdlàg paths" seems a bit convoluted for what the authors have in mind
- L 129 and 216: Lokka et al (2004) do not consider multiplicative Lévy noise laws but a linear SPDE (Poisson equation).
- L160: $m$ is not defined (nor is $\sigma$ … but one can guess for the latter)
- L 170: $T_m$ is not defined
- L 231: the Lévy-Itô decomposition is not fully introduced in the Appendix A and it does not help to understand Eq. 12
- L 232 $\Psi(t)$ is a Green's function for physicists

---

## Author Comment (AC1)

**Reply to Reviewer 1**

We would like to express gratitude to the Reviewer for their positive evaluation of the manuscript and useful suggestions for its improvement.

**Question 1**: Page 11: In the left of Eq.(15), would it be better if $x_M(t)$ is changed to $T_m(t)$?

**Answer**: Done as suggested.

**Question 2**: The first paragraph of Page 12, the expression of set alpha better be $\alpha = \{0.5, 1, 1.5, 2\}$, not open set ( )?

**Answer**: Done as suggested.

We remark that in this version of the paper we have extended the discussion relative to the properties of noise-induced transitions driven by Lévy noise by discussing the possible origin of the power law behaviour for the expectation value of the escape times (see page 10 of the revised text) and proposing an interpretation for the geometrical properties of the escape paths based on the consideration of singular perturbations (see Section 4, pages 18-20 of the revised text). We have also extended substantially our description of the main properties of Lévy processes (see much enlarged Appendix A, pages 22-23).

---

## Author Comment (AC2)

**Reply to Reviewer 2**

We would like to express our gratitude to the Reviewer for their detailed and precise analysis of our manuscript that contributes towards its improvement. We have taken all of the comments into account in the revision, as indicated in the answers below.

**Question 1**: The introduction of this model, with the help of Eqs. 1-6 and its 9 parameters and 6 empirical functions (of location x), is quite abrupt whereas it is later summarised in three physical terms ($D_I - D_{III}$), in particular in Eq. 10. It would be useful to proceed in the opposite direction and address rather usual questions such as the sensitivity of the model to the details of these terms, and in particular to the values of their parameters, as well as the uncertainty on the empirical estimates. These questions are particularly important regarding the scaling law obtained, e.g., its robustness.

**Answer**: We agree with the reviewer that a general, compact introduction to the Ghil-Sellers model along the lines of Eq. (10) should come first. We have modified the way the model is introduced. We have also explicitly mentioned that whereas we use a specific EBM, many models derived from the Budyko and the Sellers model have a very similar behaviour, as discussed extensively in the geophysical (Ghil 1981, North et al 1981, North 1990, North and Stevens 2006) and in the mathematical literature (Hetzer 1990, Diaz et al. 1997, Kaper and Engler 2013, Bensid and Diaz 2019).

**Question 2**: The introduction of the $\alpha$-stable Lévy noises is a bit surprising and disappointing. … Two main geophysical applications of the $\alpha$-stable Lévy noises were forgotten: 1) The multiplicative cascades generated by $\alpha$-stable Lévy noises, often called "universal multifractal", have not only been widely used in geophysics but have been inspired by them (Schertzer and Lovejoy, 1987), specifically to analyse and simulate their ubiquitous intermittency and heavy tailed statistics, including at climate scales. This origin has been recognised by mathematicians who have used the term "Lévy multiplicative chaos" to emphasise their generality (e.g. Fan, 1987). 2) The fractional Fokker-Planck equations for nonlinear SDE forced by non-Gaussian stable Lévy noises (e.g. Schertzer et al (2001) and references therein) used to analyse and simulate diffusion.

**Answer**: The reviewer is right as we had missed some important geophysical studies where Lévy processes have been first discussed. This has been amended; see added text and references in the fourth paragraph of Section 1.2, page 4 of the revised text. On top of the references suggested by the reviewer, we have found of great interest recent studies by Rydin Gorjão et al. (Entropy 2021, ESD 2021, now cited), and by Li and Duan (J. Stat. Phys. 2022, now cited and commented in the Conclusions) where the Moyal-Kramers equation is considered.

**Question 3**: The authors argue for an $\alpha$-stable Lévy forcing by referring to the paleoclimatic records exhibiting strong non-Gaussian behaviour. It may be worth mentioning though that these observed heavy-tailed distributions generally do not support a power law exponent $\alpha < 2$, but a larger one that can be deduced from (i), see e.g. Schmitt et. al. (1995).

**Answer**: We thank the reviewer for pointing us to this reference, which is now cited in the text.

**Question 4**: It is also surprising that the fundamental and common property of $\alpha$-stable Lévy ($0 < \alpha < 2$) and Gaussian ($\alpha = 2$) noises to be both stable (with a precise stability meaning of the index $\alpha$) and attractive (often presented like the generalised central limit theorem) is not presented.

**Answer**: We agree with the reviewer that this in a important property that should be mentioned. We have amended accordingly the text, see in the Appendix A the paragraph after formula A2.

**Question 5**: The fundamental statistical difference between $\alpha$-stable Lévy and Gaussian extremes is nor clearly presented, although indirectly evoked: namely $\alpha$-stable Lévy have heavy tails and ($\alpha < 2$) is then both the critical exponent of the divergence of statistical moments and the exponent of these power-law tails, whereas Gaussian noises do not have these extreme behaviours.

**Answer**: We agree with the reviewer that this aspect of the properties of Lévy distribution should be more clearly stated. We have amended accordingly the text, see in the Appendix A the paragraph after formula A4.

**Question 6**: The scaling law of the increments (nonetheless used in Eq.16) and that of the Lévy jump measure are not mentioned (only indirectly in Eq. A7), whereas they play a key role.

**Answer**: We agree with the reviewer that this aspect of the properties of Lévy distribution should be more clearly stated. We have amended accordingly the text, see in the Appendix A the paragraph after formula A2 and the paragraph that includes formula A10.

**Question 7**: Only symmetric $\alpha$-stable Lévy noises are introduced, whereas unlike Gaussian noises, $\alpha$-stable Lévy noises are easily skewed and only extremely skewed $\alpha$-stable Lévy noises can generate multiplicative cascades (i) with finite (theoretical) statistics and avoid spurious (empirical) estimates. The latter may impact the rationale (L 309-312) to limit the range of intensity $\varepsilon$.

**Answer**: We have decided to consider symmetric $\alpha$-stable Lévy noises because we wanted to have a simple mathematical model allowing for jumps in both directions and we wanted to be able to follow, at least heuristically, the mathematical theory developed by Imkeller and collaborators. Having a strongly skewed process would have made it very hard to explore the full phase space (a prescribed skewness would favour either the W->SB transition or the SB->W transition) and to compare the results with the Gaussian case, which is an important aspect of our work, see comment at page 9. Nonetheless, we have now added some information on the skewed case.

**Question 8**: It seems that Appendix A is unbalanced mainly recalling properties of Lévy processes instead of the specific properties of the $\alpha$-stable Lévy processes. On the contrary, the physical properties of the latter recapitulated above may help to answer the question: is there a physical reason to escape the "Gaussian rigidity" and to consider $\alpha$-stable Lévy fluctuations?

**Answer**: In the revised version of the manuscript namely in Appendix A we have better highlighted the specific properties of $\alpha-$stable Lévy processes. See above for our consideration of the physical meaning of the experiments we have performed.

**Question 9**: It is necessary to discuss the robustness of the result, especially because of the many parameters and empirical functions involved in the climate model. Since the numerically observed scaling corresponds to that of an additive stochastic forcing, it is suggested to first assess the multiplicative character, i.e., whether the stochastic forcing in such a regime or only very weakly multiplicative. This depends exclusively on the effective temperature dependence of the albedo term (Eq. 5). It could be then important to estimate the latter.

**Answer**: We understand the point made by the reviewer but here we disagree with them because we have considered an extremely well studied energy balance model. We have used standard parameter choice that support the existence of a broad region in parameters' space where multistability is observed. It is not here our goal to study the sensitivity of the properties of the Ghil-Sellers model to its parameters; this would pertain to another study. As of the multiplicative nature of the noise, the albedo depends very strongly on the temperature. Hence, the multiplicative nature of the noise is relevant. Note that the Ghil-Sellers model is constructed in such a way that such a dependence is of key importance. In the present version of the manuscript we have proposed an interpretation to clarify how the power law dependence of the expectation of the escape times emerges (page 11 of the new manuscript).

**Question 10**: Despite the initially strong physical orientation of the manuscript, the authors do not hesitate to finally abandon some physical relevance, accepting negative values of the solar irradiance, "to be able to stick to the desired mathematical framework" (L 314-316), which is in turn substituted by a numerical framework with limited ranges of noise intensity $\varepsilon$. These can contribute to overlooking the fact that symmetric $\alpha$-stable Lévy noises generate much more frequent negative fluctuations than Gaussian noises and thus a significant gap between mathematical convenience and physics. The authors partly acknowledge this problem in their conclusions in terms of general heuristics (e.g., L 449-452), but they could be more precise in their critical analysis.

**Answer**: We thank the reviewer for making this point. Our investigation takes as starting point an important problem of climate dynamics and is indeed physics-motivated. Nonetheless, we are mostly interested in investigating the impact of abandoning

the standard setting of Gaussian noise law. Using Lévy noise leads unavoidably to experience occurrences of (unphysical) negative values for the solar irradiance. In the revised version of the manuscript, we have better clarified the interplay between mathematical framework and physical relevance in our study; see discussion at the end of Sect.3, of Sect. 4, and Appendix C.

**Question 11**: Minor comments **L 124**: the expression "discontinuous càdlàg paths" seems a bit convoluted for what the authors have in mind; **L 129 and 216**: Lokka et al (2004) do not consider multiplicative Lévy noise laws but a linear SPDE (Poisson equation); **L160**: $m$ is not defined (nor is $\sigma$ ... but one can guess for the latter); **L 170**: $T_m$ is not defined; **L 231**: the Lévy-Itô decomposition is not fully introduced in the Appendix A and it does not help to understand Eq. 12; **L 232**: $\Psi(t)$ is a Green's function for physicists.

**Answer**: All amendments proposed by the reviewer are included in the text on pages 5, 7, 9 and 24.